# Deep molecular profiling of synovial biopsies in the STRAP trial identifies signatures predictive of treatment response to biologic therapies in rheumatoid arthritis

Myles J. Lewis [1,2,37] ✉, Cankut Çubuk [1,2], Anna E. A. Surace [1,2], Elisabetta Sciacca [1,2], Rachel Lau [1,2], Katriona Goldmann[1,3], Giovanni Giorli[1], Liliane Fossati-Jimack [1,2], Alessandra Nerviani [1,2], Felice Rivellese [1,2], Costantino Pitzalis [1,2,4,37] ✉ & the STRAP collaborative group*

Approximately 40% of patients with rheumatoid arthritis do not respond to individual biologic therapies, while biomarkers predictive of treatment response are lacking. Here we analyse RNA-sequencing (RNA-Seq) of pre-treatment synovial tissue from the biopsy-based, precision-medicine STRAP trial (*n* = 208), to identify gene response signatures to the randomised therapies: etanercept (TNF-inhibitor), tocilizumab (interleukin-6 receptor inhibitor) and rituximab (anti-CD20 B-cell depleting antibody). Machine learning models applied to RNA-Seq predict clinical response to etanercept, tocilizumab and rituximab at the 16-week primary endpoint with area under receiver operating characteristic curve (AUC) values of 0.763, 0.748 and 0.754 respectively (*n* = 67-72) as determined by repeated nested cross-validation. Prediction models for tocilizumab and rituximab are validated in an independent cohort (R4RA): AUC 0.713 and 0.786 respectively (*n* = 65-68). Predictive signatures are converted for use with a custom synovium-specific 524-gene nCounter panel and retested on synovial biopsy RNA from STRAP patients, demonstrating accurate prediction of treatment response (AUC 0.82-0.87). The converted models are combined into a unified clinical decision algorithm that has the potential to transform future clinical practice by assisting the selection of biologic therapies.

Over the past 20 years, targeted biological disease-modifying anti-rheumatic drugs (b-DMARD) have led to major improvements in the outlook of patients with rheumatoid arthritis (RA). However, individual bDMARDs have an ~40% failure rate, while 5–20% of patients are refractory to current medications[1,2]. Thus, predicting whether a patient

will respond to a particular therapeutic agent remains a major goal. Previous small observational studies in blood and synovium have reported a number of possible biomarkers predictive of response to TNF inhibitors (reviewed in refs. 3,4). Some of the stronger candidates include serum measurement of the B-cell attracting chemokine

[1]Centre for Experimental Medicine & Rheumatology, William Harvey Research Institute, Barts and The London School of Medicine and Dentistry, Queen Mary University of London, London, UK. [2]Barts Health NHS Trust and National Institute for Health and Care Research (NIHR) Barts Biomedical Research Centre (BRC), London, UK. [3]Alan Turing Institute, London, UK. [4]IRCCS Istituto Clinico Humanitas Rozzano (MI), Milan, Italy. [37]These authors jointly supervised this work: Myles J. Lewis, Costantino Pitzalis. *A list of authors and their affiliations appears at the end of the paper. ✉e-mail: myles.lewis@qmul.ac.uk; c.pitzalis@qmul.ac.uk

CXCL13[5], which is implicated in synovial lymphoid neogenesis[6], and the myeloid marker complex serum calprotectin[7]. Potential predictors of response to the anti-CD20 B-cell depleting agent rituximab include specific blood memory B-cell and plasmablast sub-populations, immunoglobulin J-chain mRNA levels (IgJ) and Fc gamma receptor IIIa levels[8–10], whole blood transcriptomic signatures including low levels of interferon response genes[11,12], synovial gene expression profiles[13], and serum cytokines including IL-33 and CCL19[14,15]. Putative biomarkers predictive of response to IL6 pathway inhibitors include genetic markers such as SNPs at the *IL6R* locus, transcriptomic signatures again including blood interferon response genes, serum markers including cytokines IL6, IL-8 and the adhesion molecule soluble ICAM-1[5], and cellular biomarkers such as NK cells (reviewed in ref. [16]). However as yet none of these proposed biomarkers have been successfully translated to clinical grade assays for stratifying individuals to different treatments.

Major progress in this area was made through the R4RA trial, a biopsy-driven, open-label randomised controlled trial (RCT) in which patients underwent ultrasound-guided synovial biopsy and, following stratification into B-cell-rich and B-cell-poor through a CD20 immunohistochemical score, were randomised to either rituximab or tocilizumab[17]. R4RA tested the hypothesis that in patients stratified for low/absent synovial CD20+ B-cells (the target for rituximab), tocilizumab, a specific IL6-receptor inhibitor, would be superior. Of relevance, when using the histological CD20 B-cell-poor classification, the primary endpoint (CDAI ≥50% improvement) was not met.

However, when patients were classified as B-cell poor using a validated RNA-Seq based biomarker panel of 73 B-cell-specific genes[18], both the primary endpoint (CDAI ≥50%) and low disease activity (CDAI <10.1) reached statistical significance, while machine learning modelling of synovial biopsy RNA-Seq data identified signatures predictive of response to rituximab and tocilizumab treatment[19].

Here, following the same strategy, we applied deep molecular phenotyping and machine learning modelling to synovial RNA-seq from patients included in the recently published biopsy-driven, stratified-medicine STRAP trial[20]. We identified differentially expressed genes between responders and non-responders and examined the relationship between single-cell RNA-seq[21,22], cell subset modules and response to each drug using deconvolution. Additionally, we defined predictive RNA-seq-based models of response for the three randomised drugs in STRAP, etanercept, tocilizumab and rituximab[20], that, when converted into nanoString panels tested in available synovial RNA from STRAP patients, could accurately predict actual observed response in 79–85% of patients (AUC 0.82–0.87).

## Results

### Differential gene expression analysis of synovial tissue RNA-seq identifies signatures of responsiveness to etanercept, tocilizumab and rituximab

About 223 patients were included in the primary analysis of the STRAP trial[20], of which 208 had post-quality control RNA-Seq data from synovial biopsies; of these, 67 had been randomised to etanercept, 69 to tocilizumab and 72 to rituximab. Baseline characteristics, disease activity, and synovial B cell group are reported in Supplementary Table 1. Patients were assessed for clinical response 16 weeks after starting treatment using the ACR20 criteria (the original primary endpoint measure), with 38 (57%) responding to etanercept, 51 (74%) responding to tocilizumab and 44 (61%) responding to rituximab. Differentially Expressed Gene (DEG) analysis using DESeq2 identified 44 genes differentially expressed between responders and non-responders to etanercept, 90 genes in tocilizumab and 44 genes for rituximab response (FDR <0.05, Supplementary Data 1) in baseline synovial biopsies (Fig. 1a, c, e). Etanercept and rituximab response were associated with increased expression of B-cell genes, including

immunoglobulin chain genes (*IGHD*, *IGKV1-37*), and B-cell surface receptors *MS4A1* (CD20), *CD22*, BAFF receptor (*TNFRSF13C*) and B-cell differentiation genes (*BLK*, *PAX5*). Tocilizumab response was associated with upregulation of the acute-phase reactant *SAA2* (serum amyloid A2), while tocilizumab non-response genes included *IL18RAP*, which has been previously linked to therapeutic response in RA[23]. Collagen genes (*COL23A1*, *COL11A2*) and matrix metalloproteinase 9 (*MMP9*), consistent with tissue remodelling, were associated with non-response to both etanercept and rituximab. Differential expression analysis was repeated with adjustment for biopsy joint size as a covariate, but this did not significantly alter or improve the overall analysis (Supplementary Fig. 2).

QuSAGE modular analysis of differentially expressed genes showed an increase in multiple B cell modules in responders to etanercept, which were the only *q* value significant modules for this drug. (Fig. 1b and Supplementary Data 2). For tocilizumab, on the other hand, the same S46 B cell module was associated with tocilizumab non-response (Fig. 1d). Dendritic cells and interferon alpha modules were upregulated in responders to tocilizumab. For rituximab, several modules related to B cells, T peripheral helper cells (Tph), and NK and T-cells were increased in responders, while fibroblast-associated modules were associated with non-response (Fig. 1f).

### Defining shared and differential gene signatures of response/resistance among etanercept, tocilizumab and rituximab

To investigate common molecular patterns of response/resistance to treatment, we conducted a drug-independent analysis comparing responders (*n* = 133) and non-responders (*n* = 75) in the whole cohort. DEG analysis (Fig. 2a and Supplementary Data 3) showed 20 upregulated genes (FDR <0.05) in the responder group that were associated with inflammation, immunoregulatory interactions and B-cell proliferation such as the Fc receptor-like 1 and 2 (*FCRL1-2*), BAFF receptor (*TNFRSF13C*), B lymphocyte kinase (*BLK*), interleukin 9 receptor (*IL9R*), *CD22* and *PAX5*. This is confirmed by the pathway enrichment analysis (Fig. 2b and Supplementary Data 3). Genes upregulated in the non-responder group were more abundant and heterogeneous (75 genes at FDR <0.05). Some of them are linked to altered fatty acid metabolism (e.g. *SCD* and *FASN*), others are pro-fibrotic genes such as *SCN7A*, *CNN1* and *MMP9* (Fig. 2a, b). A protein phosphatase catalytic subunit (PPP) was also found to be upregulated in this group (*PPP1R14A*), as well as Proenkephalin (*PENK*), which plays an important role in the modulation of pain perception.

The heterogeneity of resistance mechanisms found in the DEG analysis led to a separate analysis aimed at identifying and comparing the different molecular signatures driving resistance to each drug. To analyse specific differences between drugs, genes associated with lack of response to each drug were visualised with a three-way polar plot (Fig. 2c). This revealed nine genes specifically upregulated in non-responders to etanercept, 22 in rituximab and 30 in tocilizumab, respectively. Drug-specific responder genes are shown in Fig. 2d, revealing greater abundance of genes along the tocilizumab responder axis (59 genes), and fewer genes for the rituximab-specific and etanercept-specific groups (8 and 24 genes, respectively). These comparisons highlighted the differential impact of immune-related genes like *CR2*, *LTF* and *TCL1A*, all significantly upregulated in tocilizumab non-responders and in rituximab responders, with *CR2* and *TCL1A* also upregulated in the etanercept responder group (Fig. 2c–e). High levels of B-cell genes (*MS4A1*, *PAX5*, *CR2*) were significantly associated with response to etanercept and rituximab (Fig. 2e), while being upregulated in non-responders for the tocilizumab group. On the opposite trend, the keratin gene *KRT10* was upregulated in tocilizumab and rituximab responders, while higher expressions could be observed in non-responders for the etanercept group. A similar trend could be observed for *MMP9* (Fig. 2e).

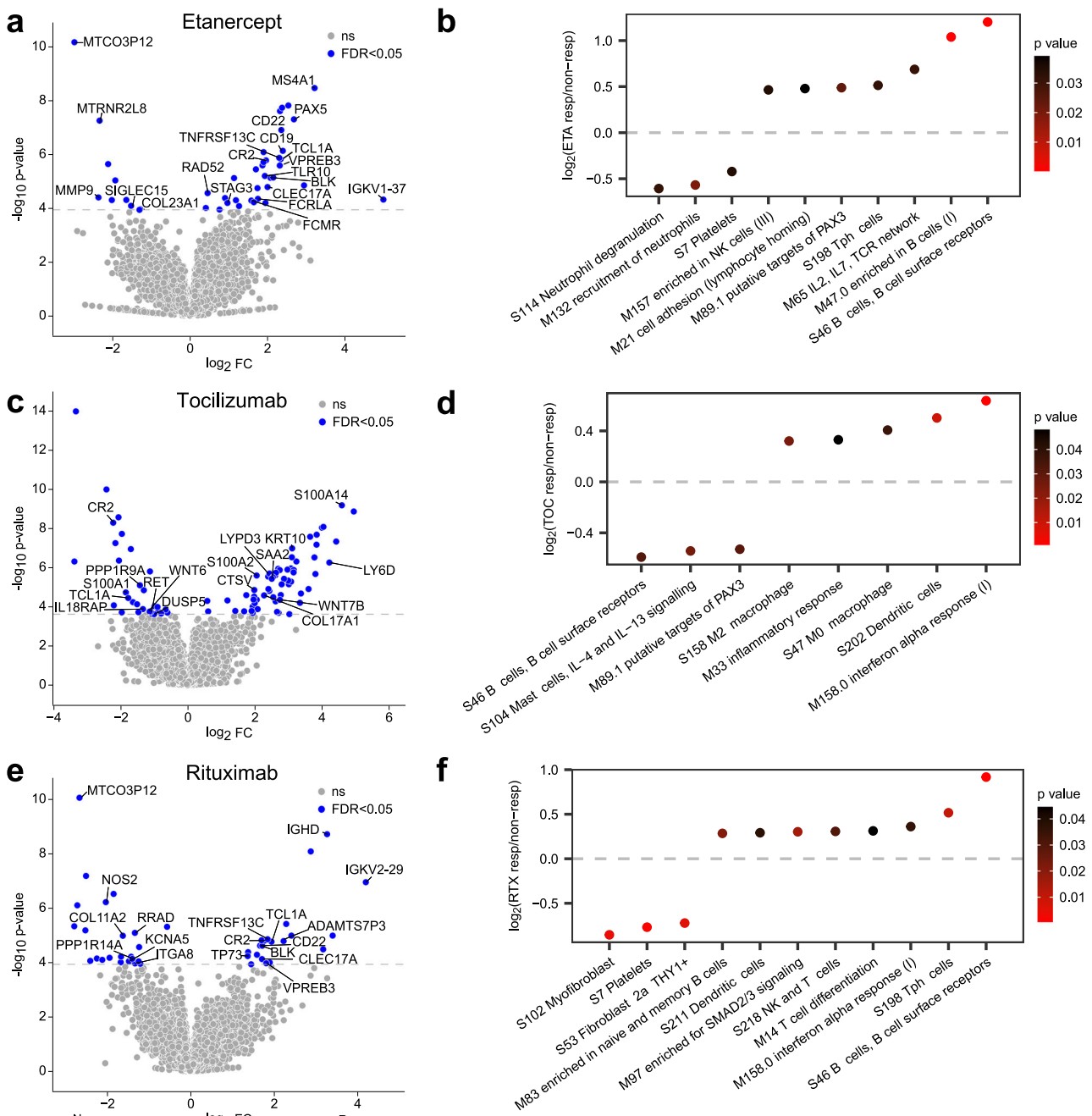

**Fig. 1 | Synovial signature of response to biologics at baseline. a**, **c**, **e** Volcano plots of differentially expressed genes from DESeq2 analysis of RNA-sequencing of baseline synovial biopsies of rheumatoid arthritis individuals receiving treatment with **a** etanercept (n = 67), **c** tocilizumab (n = 69) or **e** rituximab (n = 72) comparing ACR20 responders vs non-responders at the 16-week primary endpoint. DESeq2 statistical analysis uses generalised linear modelling of count data using the negative binomial distribution. The model included a single covariate based on principal component analysis applied to 17 muscle tissue-specific genes (see Methods). *P* values were calculated by a two-sided Wald test with FDR correction (Storey's *q* value) for multiple testing. Genes in blue are significant at FDR <0.05, genes in grey are non-significant. **b**, **d**, **f** Modular analysis applying QuSAGE statistical testing with blood-derived gene modules (Li et al., 2014) and synovium-derived WGCNA modules for 16-week ACR20 responders versus non-responders to etanercept (**b**), tocilizumab (**d**) and rituximab (**f**). Log2-fold change of responders (positive values) and non-responders (negative values) are plotted with dots colour coded for unadjusted *p* value.

## Cellular composition determined by clustering analysis and single-cell RNA-Seq subset deconvolution is associated with different patterns of drug response

To explore the association of immune cells with response to treatment (ACR20), we used clustering analysis and a modular enrichment approach to estimate the relative abundance of cells in the synovial tissue of RA, including fibroblasts, macrophages, B-cells and T-cells based on single-cell RNA-Seq subsets identified in RA synovium by

ref. 21 Clustering of patients using their immune cell subset profiles at baseline was performed to examine the RNA-seq data for underlying structure and the presence of biological subgroups. This revealed four clusters, which partially mapped to the three pathotypes as determined by histology (Fig. 3a). The pauci-immune fibroid patients were split across two clusters, namely cluster 1, which was almost entirely composed of fibroid patients and cluster 2, which contained the remainder of fibroid patients. Cluster 3 contained a mixture of diffuse-

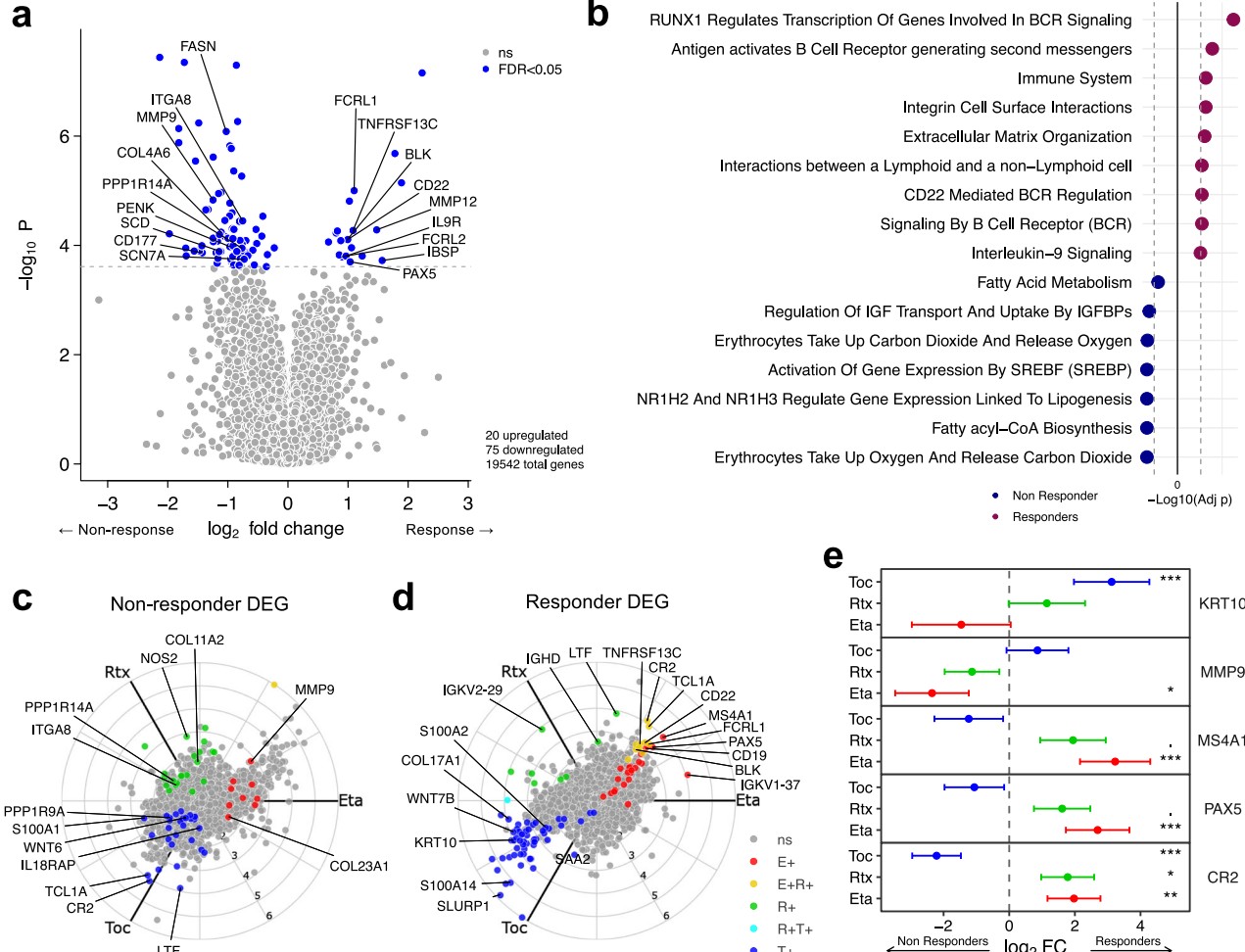

**Fig. 2 | Analysis of common and differential molecular signatures of responsiveness/resistance to etanercept, tocilizumab and rituximab. a** Volcano plot showing differentially expressed genes between all ACR20 responders ($n = 133$) and non-responder ($n = 75$) patients to tocilizumab, etanercept and rituximab combined following 16 weeks of treatment. Statistical analysis by negative binomial distribution, generalised linear regression of count data via DESeq2. *P* values were calculated by a two-sided Wald test with FDR correction (Storey's *q* value) for multiple testing. **b** Shared enriched pathways across tocilizumab/etanercept/rituximab responders and non-responders. Grey dashed lines indicate *p*-value cutoff for FDR <0.05. **c** Three-way polar plot comparing genes associated with resistance to each individual drug. Red genes ($n = 9$) are significantly upregulated only in non-responder patients treated with etanercept. Genes in green ($n = 22$) are significantly upregulated in rituximab non-responders only. Blue genes ($n = 30$) are significantly upregulated in the tocilizumab non-responder group. **d** Three-way polar plot of significantly upregulated genes in responder patients to etanercept (24 genes, red dots), rituximab (eight genes, green dots) and tocilizumab (59 genes, blue dots). **e** Forest plots of individual genes showing different log2FC (responders/non-responders) in each drug. ***$p < 0.001$, **$p < 0.01$, *$p < 0.05$, '.' $p < 0.10$ using FDR-adjusted two-sided Wald test *p* values. Precise *p* values are available in the supplementary material. Error bars show 95% confidence intervals.

myeloid patients and lympho-myeloid patients, while Cluster-4 was almost entirely composed of lympho-myeloid patients with the highest levels of B/T-cells by CD20/CD3 histology score and plasma cells by CD138 score. Estimated abundance of DKK3+ fibroblasts (SC-F3) and CD34+ sublining fibroblasts (SC-F1) delineated cluster 1, while CD55+ lining fibroblasts and NUPR1+ macrophages defined cluster 2, the other fibroid-associated cluster, with intermediate levels of both cells in cluster-3. This suggests that specific synovial single cell subtypes can define additional subgroups into which the original three pathotypes[18,24] can be further subdivided, consistent with the most recently published study from the AMP Consortium[22].

These findings allowed us to conduct the following analyses by either combining all samples or analysing separately for each treatment. Differential abundance of predicted synovial single cell subtypes between responders and non-responders to any of the drugs combined (Fig. 3b) was compared with differential abundance to each drug specifically (Fig. 3c). This showed that HLA-DRA^high sublining fibroblasts (SC-F2), a pro-inflammatory subset associated with leucocyte-rich synovial infiltration in RA, was significantly higher ($P_{F2\_any} = 0.008$) in

responders in the any treatment comparison, which is in parallel to previous findings reported earlier in the R4RA cohort[19]. In R4RA, DKK3+ sublining fibroblasts (SC-F3) were increased in patients refractory to treatment. Here, DKK3+ sublining fibroblasts (SC-F3) showed a trend to a non-significant increase in etanercept and rituximab-treated patients, while CD34+ sublining fibroblasts (SC-F1) were significantly increased in non-responders to etanercept.

Patients treated with rituximab did not show significant changes in the populations of monocyte/macrophage subsets; however, those who responded to etanercept had a significantly higher proportion of estimated IL1B+ pro-inflammatory macrophages (SC-M1) compared to non-responders, and tocilizumab responders had a higher proportion of NUPR1+ and IFN-activated macrophages (SC-M2 and SC-M4). The signatures for predicted IGHD+CD27− naive (SC-B1) and IGHG3+CD27+ memory (SC-B2) B-cell subsets were significantly higher in responders who received etanercept and any treatment at baseline ($P_{B1\_eta} = 0.036$, $P_{B1\_any} = 0.018$, $P_{B2\_eta} = 0.032$ and $P_{B2\_any} = 0.004$). The Tph subset (SC-T3) showed one of the greatest fold change among all immune cell subsets and was significantly upregulated in the three responder

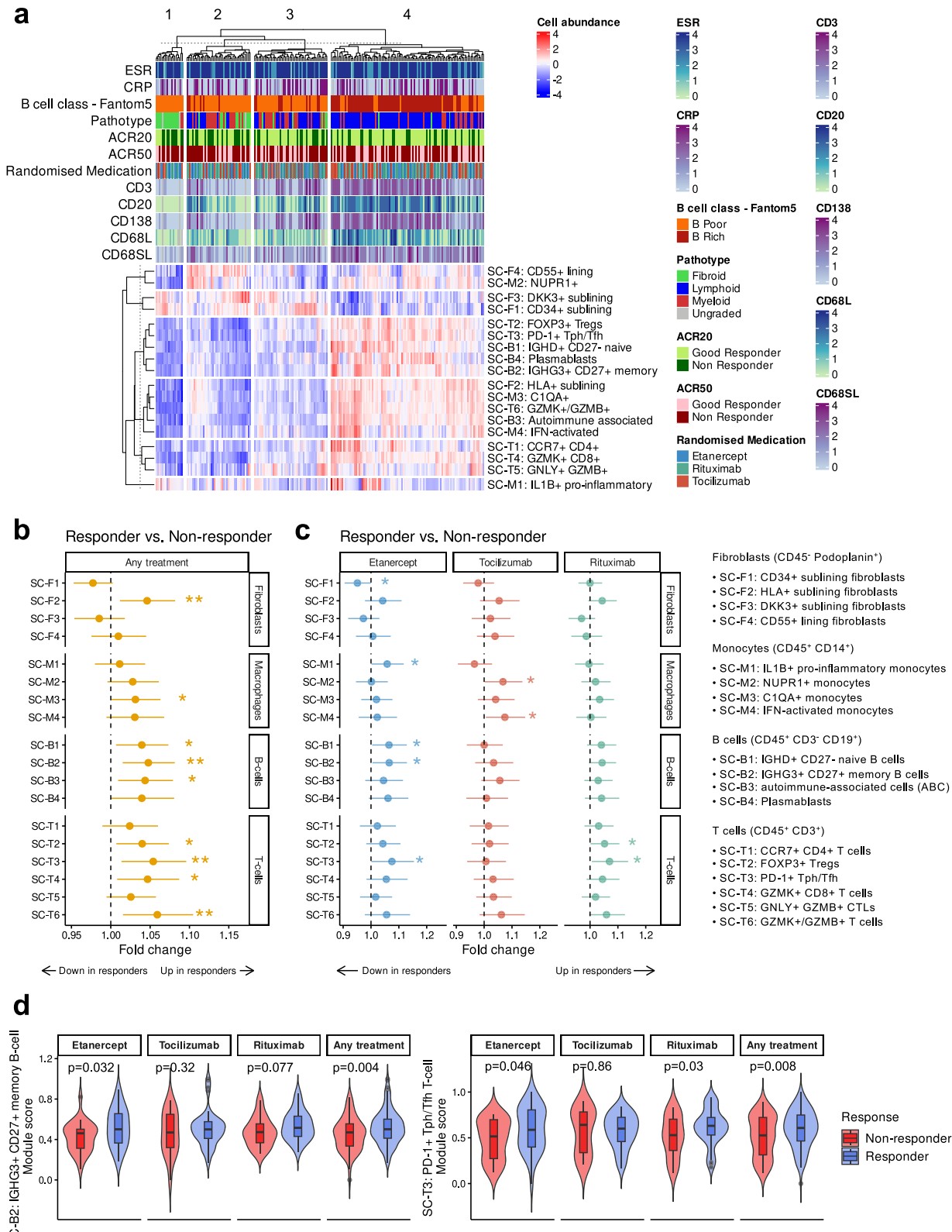

**Figure caption area**

groups: etanercept, rituximab and any treatment (Fig. 3c, d). A significant increase in estimated FOXP3+ regulatory T-cells (SC-T2) was seen in the any treatment comparison and in rituximab responders specifically. Similarly, SC-T4 (GZMK+CD8+ T-cells) and SC-T6 (GZMK+/GZMB+ T-cells) were predicted to be increased in responders in any treatment comparison. A three-way comparison between cell subtypes

in responders specifically across each of three treatment groups did not show significant differences between responders for each drug (Supplementary Fig. 3a). A similar analysis looking for cell subtype differences specific to non-responders also produced non-significant results for all cell subtypes and treatment pairs tested (Supplementary Fig. 3b).

**Fig. 3 | Single-cell subset patterns of responder and non-responder patients to etanercept, tocilizumab and rituximab. a** Heatmap showing estimated immune cell subset profiles of all individuals at baseline, calculated by gene module score using Seurat. Individuals (columns) were clustered using the Euclidean distance metric and complete linkage clustering method. Upper tracks show ESR, CRP, cell type (B cell rich/poor), pathotype, ACR20 and ACR50 response, randomised medication (treatment) and histological scores for CD3, CD20, CD138, CD68L (lining) and CD68SL (sublining). **b** Forest plot showing mean fold-changes of single-cell subsets that are differentially present in any responders compared to any non-responders. Error bars show 95% confidence intervals. Statistical analysis (two-sided) by linear model using limma. Significant fold-changes are indicated with asterisks (*$p < 0.05$, **$p < 0.01$, ***$p < 0.001$). Precise $p$ values are available in the supplementary material. **c** Forest plot showing fold-changes of single-cell subsets that are differentially present in responders compared to non-responders separately in each medication. **d** Box plots showing module scores of SC-B2 (IGHG3$^+$CD27$^+$ memory B-cell), SC-T3 (PD-1$^+$ Tph/Tfh) T-cell subsets for etanercept, rituximab and any treatment (either etanercept, rituximab or tocilizumab) groups. Box plots show median, upper and lower quartiles, with whiskers denoting maximal and minimal data within 1.5 × interquartile range.

## Unbiased clustering of synovial RNA-seq in STRAP patients defines three distinct molecular groups, confirmed in the independent R4RA biopsy cohort

To establish an unbiased definition of gene expressions at baseline, we plotted a heatmap of the most highly expressed genes ($n = 3411$), which revealed the presence of three molecular clusters (Fig. 4a and Supplementary Data 4). Patients belonging to the first cluster ($n = 37$) showed improved response to therapy according to DAS28-CRP criteria, with only one patient classified as a non-responder out of 37 ($p = 0.006$, Supplementary Table 3). The second cluster ($n = 75$) was dominated by the fibroid pathotype, with 28 fibroid samples out of 34 falling into cluster 2. Coherently, this cluster was also associated with samples histologically classified as B-cell poor and with lower levels of CD138 ($p < 0.001$, Supplementary Table 3). The third cluster ($n = 96$) showed the highest percentage of rheumatoid factor (RF) and anti-citrullinated protein autoantibodies (ACPA) positive samples ($p = 0.004$, Supplementary Table 3).

To investigate the presence of these clusters in an independent cohort, we generated an analogous heatmap of the most highly expressed genes ($n = 2259$) in the baseline population of the R4RA clinical trial (Fig. 4b)[17,19]. Interestingly, the resulting heatmap revealed three unsupervised clusters whose genes significantly overlapped with the corresponding clusters found in STRAP (Fig. 4d). Building on these findings, we conducted a functional enrichment analysis to assess the role of the genes found in each cluster. The first cluster (983 genes) is mostly associated with L13a-mediated silencing of ceruloplasmin and with regulation of expression of SLITs And ROBOs (Fig. 4c and Supplementary Data 4). Ceruloplasmin levels are high during inflammation, and its silencing is typically initiated by the cytokine interferon-gamma (IFN-γ), which triggers a signalling cascade that leads to the phosphorylation of the ribosomal protein L13a. Phosphorylated L13a then dissociates from the 60S ribosomal subunit. The SLIT-ROBO pathway is involved in regulating inflammatory responses and neuronal axon guidance. For example, *SLIT2* has been found to have anti-inflammatory properties, inhibiting leucocyte chemotaxis[25], while Denk et al suggested that *SLIT3* might have a protective role in RA by limiting synovial fibroblast invasion[26]. Other pathways associated with this cluster are Axon Guidance, which is known to be linked to SLIT-ROBO signalling, cellular response to hypoxia, and negative regulation of NOTCH4 Signalling. Hypoxia often characterises the RA joint microenvironment and can regulate the expression of Notch receptors, including NOTCH4[27]. Its negative regulation suggests mitigation of its pro-angiogenic role[28]. However, NOTCH1 and especially NOTCH3 have been more heavily implicated in pathogenic RA fibroblast signalling[29]. Pathways associated with cluster 2 (1420 genes) include multiple extracellular matrix related pathways highly correlated with fibroblasts activation (Fig. 4c). The third cluster (1008 genes) showed clear molecular signature of inflammation driven by cytokines (Fig. 4c) with neutrophil degranulation, cytokine signalling and signalling by interleukins being the most upregulated pathways.

Finally, to compare these molecular groups with the histologically defined pathotypes we plotted a principal component analysis (PCA) colour coded by pathotype first, and by the new molecular groups later (Fig. 4e). The resulting plots highlighted overlap of the second molecular cluster with the fibroid pathotype, as already observed in the heatmap (Fig. 4a), while lymphoid and myeloid samples were present in both cluster 1 and 3, suggesting additional heterogeneity within these pathotypes.

## Machine learning models define synovial signatures predictive of drug response to etanercept, rituximab and tocilizumab

Machine learning (ML) predictive models were constructed to establish the ability of baseline synovial tissue gene expression and clinical parameters to predict treatment response. In order to develop a predictive test to be used in clinical practice, we aimed to convert RNA-Seq signatures to an nCounter-based assay. Models were built from baseline clinical parameters and RNA-Seq data restricted to a synovial specific 524-gene nCounter panel (507 target genes, 17 housekeeping genes), custom-made in collaboration with nanoString, covering genes linked to synovial biology, pathotypes, RA pathogenesis and response gene signatures from previous RNA-Seq studies[18,19,24,30] through a three-stage design process (summarised in Supplementary Fig. 4).

Performance was tested in an unbiased manner, maintaining separation of samples for training and testing using a machine learning pipeline (Fig. 5a) involving 10 × 10-fold nested cross-validation with 25 repeats[31]. Eight machine learning model types were tested (see Methods). Predictive performance was measured by AUC (area under ROC curve) for response to each drug, comparing the primary endpoint ACR20 response against five other response endpoints (Supplementary Figs. 5–7). This showed that for etanercept and tocilizumab, the best prediction of response was seen for target DAS28-ESR (<3.2), which was thus selected for model optimisation. DAS28 has a major advantage over ACR20/50/70 response criteria in that DAS28 only requires 28 joints to be assessed by clinicians, whereas the ACR criteria require a total of 68 joints to be assessed. Although the ACR criteria are a gold standard for use in RCT, DAS28 is in widespread use and can be assessed in routine clinical practice. Therefore, we chose to train models to predict a DAS28-based outcome as these would be more readily validated in future. Due to difficulties with training a binary response model for rituximab, ordinal regression on DAS28-ESR response categorised into four levels was used for model training (see Methods), followed by converting the fitted regression model to a binary response for final performance estimation by AUC.

For etanercept, the best model system was an elastic net regression model (glmnet) (Fig. 5b), with a final 19-parameter model of 17 genes and two clinical parameters with nested CV AUC of 0.763 (Fig. 5c). For tocilizumab the final model was a 28-parameter gradient boosted machine model (gbm) using 26 genes and two clinical parameters with a nested CV AUC of 0.748. The final rituximab extreme gradient boosting model (xgbLinear) contained three clinical parameters and 25 genes, leading to a nested CV AUC of 0.754. All three models showed reasonable accuracy (range 71.6 to 75.4%) and balanced accuracy (70.2 to 71.6%) (Supplementary Tables 4, 5). In comparison, anti-CCP and rheumatoid factor (RF) titre as continuous variables were relatively poor predictors of response to each of the three drugs in STRAP and the two drugs (tocilizumab and rituximab)

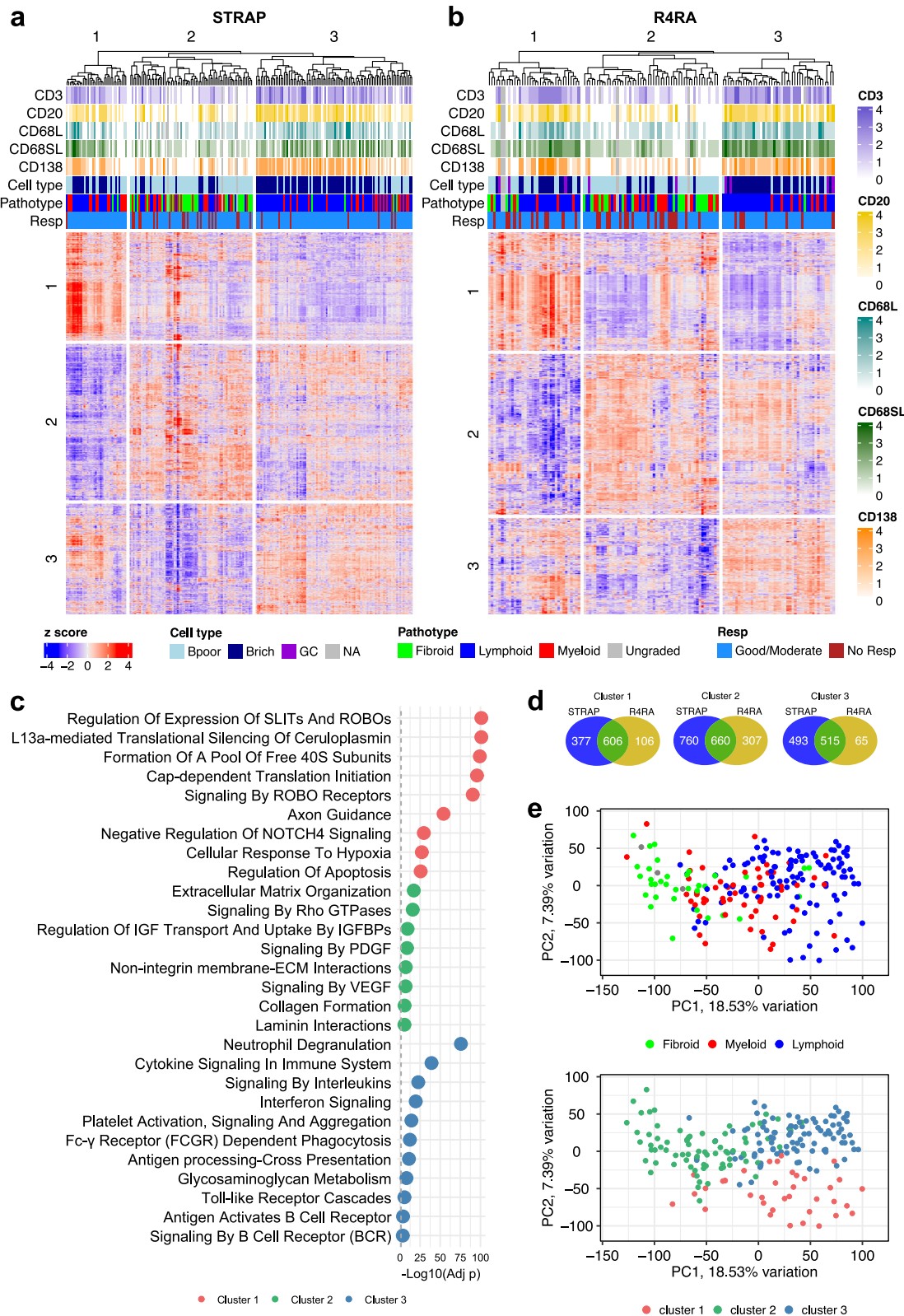

used in R4RA (Supplementary Fig. 8). Anti-CCP/RF seropositivity (binary) showed modest associations with increased response to etanercept and rituximab in STRAP using different endpoints (Supplementary Table 8), but no associations were observed with tocilizumab. However, these associations between CCP/RF seropositivity and response endpoints disappeared when STRAP and R4RA were combined (Supplementary Table 8).

To interpret the models, variables were ranked according to their importance in models which was averaged across the outer CV folds and the final model to give a reliable estimate of variable importance and its variance (Fig. 5d). These plots also demonstrate how many times each gene was selected across the outer CV folds, showing how stable usage of each gene was across repeated models. This showed that the most important genes were consistently utilised in models

**Fig. 4 | Unsupervised clustering reveals molecular groups of patients that are reflected in an independent cohort. a** Unsupervised k-means clustering on the 3411 most expressed genes from all baseline samples (n = 208) reveals three distinct subgroups of patients. Upper tracks show histological scores for CD3, CD20, CD68L, CD68SL, CD138, cell type (B cell rich/poor), pathotype and DAS28 CRP response. **b** Unsupervised k-means clustering on the 2259 most expressed genes from all baseline samples (n = 133) of the R4RA cohort reveals subgroups of patients that share common molecular signatures with the clusters found in the STRAP cohort. Upper tracks show histological scores for CD3, CD20, CD68L, CD68SL, CD138, cell type (B cell rich/poor), pathotype and DAS28 CRP response. **c** Pathway analysis of the three gene clusters identified in the STRAP cohort (cluster 1 = 983 genes, cluster 2 = 1420 genes, cluster 3 = 1008 genes). **d** Venn diagrams showing numbers of distinct and shared genes in the three clusters independently obtained in STRAP and R4RA. **e** The PCA plots from baseline samples of the STRAP cohorts colour coded by pathotype (top) and unsupervised molecular clusters identified by k-means clustering (bottom).

across different outer CV folds. The aggregated ranking of variable importance values from outer CV folds, as well as the final model, showed a similar pattern of consistent ranking of variables by model importance (Supplementary Fig. 9).

### Validation of STRAP synovial signatures predictive of drug response to rituximab and tocilizumab in the independent R4RA cohort

In order to validate the machine learning models in an independent cohort, the models for rituximab and tocilizumab, which were trained on STRAP data, were applied to existing data from the R4RA trial (n = 133), where these two biologics repeated the randomised study drugs[17]. Of the original 164 patients randomised on entry to the R4RA trial, 133 patients had good-quality baseline synovial RNA-Seq post-QC[19]. Predicted response based on tocilizumab and rituximab predictive models was compared to actual response in R4RA participants treated with tocilizumab (n = 65) or rituximab (n = 68), with response defined as patients achieving low disease activity (DAS28-ESR <3.2) at the 16-week endpoint. Each model showed a prediction ability for tocilizumab response with an AUC of 0.713 and of 0.786 for rituximab response (Fig. 5e) and balanced accuracy of 0.752 and 0.680 (Supplementary Table 6), confirming validation of each of these models in an external, independent RA cohort.

Full models were also compared against models built using clinical parameters alone (Supplementary Fig. 10). These showed significantly inferior prediction for all three drugs (etanercept AUC 0.601, tocilizumab AUC 0.673, rituximab AUC 0.642) in STRAP compared to the full models including synovial gene expression. In addition, retesting of the clinical-only parameter models trained in STRAP against response to tocilizumab and rituximab in R4RA also showed poorer response prediction (tocilizumab AUC 0.624, rituximab AUC 0.653).

### Rebuilding models using pooled STRAP and R4RA data

For comparison purposes, RNA-Seq data and clinical data from patients in STRAP and R4RA treated with tocilizumab (n = 134) and rituximab (n = 140) were combined in order to build models using the largest possible datasets. Machine learning using the same repeated nested CV pipeline applied to this enlarged data resulted in an 11-parameter glmnet model for tocilizumab prediction and ten-parameter partial least-squares regression (pls) model for rituximab prediction with nested CV AUC of 0.785 and 0.750 respectively (Supplementary Figs. 11, 12), which were comparable to results obtained using STRAP alone for training with similar accuracy and balanced accuracy. Confusion matrices (Supplementary Table 7) showed that the prediction models enriched the delineation of responders and non-responders: individuals predicted to be tocilizumab responders showed an enriched response rate of 75% (61/81) compared to 32% (17/53) response in predicted non-responders. Similarly, rituximab predicted responders showed an actual response rate of 76% (16/21) compared to 18% (21/119) response in the predicted non-response group. In contrast in STRAP & R4RA combined, RF or CCP positive patients showed no difference in response rates compared to seronegative patients following treatment with tocilizumab (57% vs 61–63%) and only a moderate increase in response rate with rituximab (28–30% vs 17–19%) (Supplementary Table 8). Thus, the new prediction models provide additional benefit above and beyond simple seropositivity for predicting response to biologic therapies.

### Conversion of machine learning models into nCounter panels demonstrates high predictive value for patient stratification by response to specific biologics

RNA-Seq is a highly effective research tool, but technical and analytical barriers prevent its use in routine clinical practice, while nCounter is a rapid multiplexed gene expression assay, which has achieved FDA certification for clinical adoption in the field of breast cancer[32]. Thus, we converted the RNA-seq machine learning models into an nCounter-based system, which we validated using available RNA from synovial biopsies (n = 118) from the STRAP trial, utilising the custom 524-gene nCounter panel mentioned above. The custom nCounter panel was designed in a three-stage process, shown in Supplementary Fig. 4 and described in full in the Methods. In brief, an initial set of 798 genes linked to RA pathotypes[18], genes encoding proteins interacting with known important RA therapeutic targets (TNF/TNFR, IL6/IL6R, etc.), genes from previous RA treatment response prediction models[19,20,24,30], cell-specific genes from scRNA-seq studies[21,22,33] and synovium-specific housekeeping genes based on analysis of PEAC, R4RA and STRAP RNA-seq data were submitted for the first custom panel. This was tested on 48 synovial biopsy RNA samples. A second stage 523-gene panel was redesigned to improve background noise detected in negative control probes and tested on a further 48 samples. The finalised panel included a total of 507 target genes and 17 housekeeping genes (a total of 524 genes, Supplementary Data 5).

Comparison of nCounter normalised counts against RNA-Seq on the same samples showed strong correlation between nCounter and RNA-Seq for the vast majority of genes (Supplementary Fig. 13). To be able to input nCounter gene counts into the machine learning models constructed on the RNA-Seq data (Fig. 5c, d), we developed an algorithm to convert nCounter count data for each gene to RNA-Seq scale (pseudo-RNA-Seq), using linear models fitted for each gene. Figure 6a outlines the pipeline used to test the validity of the nCounter biomarker panel for predicting response to each of the three biologic drugs randomised in STRAP. nCounter data for each cohort (etanercept n = 39, tocilizumab n = 34, rituximab n = 45, total n = 118) was converted to pseudo-RNA-seq and inputted into the relevant ML model (described in Fig. 5c, d) for each cohort. We tested predicted response against actual response at 16 weeks (reference) to the drug received by each patient in the STRAP trial. A balanced accuracy of 0.79 was achieved for the etanercept and tocilizumab models and 0.81 for the rituximab model (Fig. 6b), while an AUC of 0.87 was observed for etanercept response, 0.82 for tocilizumab response and 0.87 for rituximab response (Fig. 6c). Thus, all three RNA-Seq derived machine learning models were successfully converted and validated using an alternative, non-sequencing-based assay. This led to the development of an algorithm for future clinical use where the nCounter biomarker panel can assign patients to either (i) a TNF-inhibitor, (ii) an IL6-inhibitor, (iii) a B-cell depleting agent, or (iv) a biomarker-negative group if patients have low probability (all <50%) of responding to all three classes of drugs (Fig. 6d).

## Discussion

We performed a comprehensive analysis of RNA-Seq data from synovial biopsies from the STRAP trial that led to the identification of gene

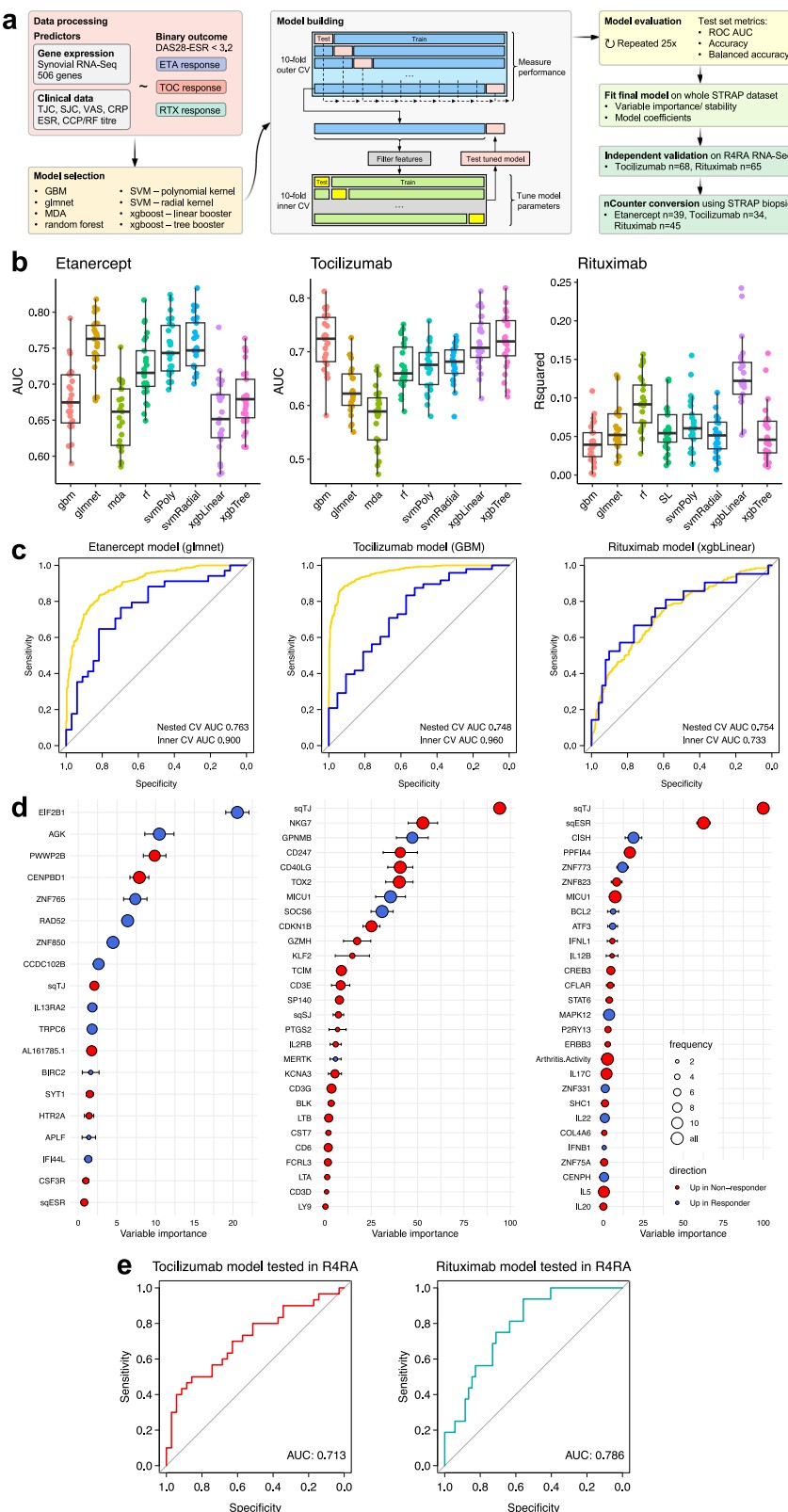

signatures associated with response to etanercept, tocilizumab and rituximab (Fig. 1a, c, e). B-cell gene modules had a strong theme predicting response to both rituximab and etanercept (Fig. 1b, f), while chemokine and cytokine gene modules were associated with response to tocilizumab (Fig. 1d). Comparative analysis identified both shared gene response and non-response signatures (Fig. 2a, b) with Fc receptor mediated inflammatory signalling as a strong underlying theme. Drug-specific response signatures were also dissected, showing that specific genes could be associated with response to one drug and non-response to another, for example selected fibroblast genes such as *KRT10* being associated with response to tocilizumab and non-response to etanercept, while B cell associated genes *MS4A1* which encodes CD20 and the B cell-specific transcription factor *PAX5* were upregulated in etanercept responders.

**Fig. 5 | Machine learning predictive models fitted using ten-by-ten-fold nested cross-validation for response to etanercept, tocilizumab and rituximab.** **a** Schema showing a machine learning pipeline. **b** Box plots of model performance for each of the three trial drugs. Multiple types of machine learning (ML) models were fitted to baseline synovial RNA-Seq gene expression data to predict response to each trial drug at the 16-week primary endpoint, with response defined as DAS28-ESR <3.2. Model types: gradient boosted machine (gbm), elastic net regression (glmnet), mixed discriminant analysis (mda), random forest (rf), support vector machine (svm) with polynomial (svmPoly) or radial (svmRadial) kernel, extreme gradient boosting (xgboost) with tree booster (xgbTree) or linear booster (xgbLinear). Unbiased model performance was determined by $10 \times 10$-fold nested cross-validation (CV) with 25 repeats (each point shows one repeat), with the area under the receiver operating characteristic (ROC) curve as performance metric for etanercept and tocilizumab. The Coefficient of determination $R^2$ was used as a performance metric for rituximab models (see Methods), which were fitted to an ordinal (four-level) response outcome, as this led to improved final binary response

prediction. Box plots show median, upper and lower quartiles, with whiskers denoting maximal and minimal data within $1.5 \times$ interquartile range (IQR). **c** ROC curves for final best models for each drug, showing nested CV ROC and ROC calculated from inner CV folds. **d** Variable importance plots showing stability of variables selected by the final ML model for each drug across nested CV. Error bars show the standard error of mean variable importance, size of points shows frequency with which each gene/predictor was selected by models during nested CV. Colour of points shows directionality of association with response: red for genes/predictors upregulated in non-response, blue for genes/predictors upregulated in response. **e** Validation of STRAP-trained tocilizumab and rituximab machine learning models in R4RA. Models for tocilizumab and rituximab shown in **c**, **d** were applied to synovial RNA-Seq and data from patients randomised to treatment with tocilizumab (*n* = 65) or rituximab (*n* = 68) in the R4RA trial. Predicted outcome was compared to the real outcome, with response defined as DAS28-ESR <3.2 at the 16-week primary endpoint of the trial. Predictive model performance was assessed by ROC AUC.

The association of increased tissue B-cells with greater response to rituximab inherently fits with the mechanism of action of rituximab, which depletes B-cells, in keeping with the previous observations in the R4RA trial[17]. There are multiple reasons why etanercept response might be affected by the level of B-cell infiltration. Synovial B cells express TNF receptor 2 (p75)[34,35], which drives lymphocyte activation and proliferation[36]. Etanercept is unique among anti-TNF inhibitors for its added ability to block the B-cell-promoting cytokine lymphotoxin-alpha[37]. Synovial B-cell infiltration is the defining feature of the 'lympho-myeloid' pathotype, which was renamed from its earlier name 'lymphoid', because of the observation that patients with this pathotype also had the most inflammatory synovial tissue containing the highest amount of macrophages[18]. Synovial macrophage infiltration has historically been associated with radiographic progression[38] as well as therapeutic response to multiple agents[33,39,40], and synovial B-cell infiltration has shown a similar association with accelerated bone erosion[20,24]. B-cell genes as a signature for anti-TNF response could be interpreted as a marker for those samples with the highest macrophage infiltration, as well as the highest levels of tissue inflammation, with the most complex inflammatory cell infiltrates with ectopic lymphoid structure formation[41]. So it is plausible that a high synovial B-cell gene signature would be a marker of enhanced response to anti-TNF therapy and etanercept specifically.

Deconvolution of the bulk RNA-Seq data using gene modules of single cell subsets from the Accelerated Medicines Partnership (AMP) consortium[21] showed that specific estimated cellular subsets were associated with response, for example SC-M2 and SC-M4 macrophage subsets were associated with tocilizumab response while SC-M1 macrophage, and SC-B1 and SC-B2 B cell subsets were associated with etanercept response (Fig. 3b, c). These results are consistent with other recent single-cell work examining the role of specific synovial macrophage subsets in maintaining or limiting remission in RA patients[33]. Single-cell subset analysis of the bulk data segregated with pathotypes (Fig. 3a), but suggested greater complexity than the original description of the pathotypes[18,24], with for example the pauci-immune fibroid pathotype being split into two clusters based on predicted cell types including DKK3+ (SC-F3) and CD34+ (SC-F1) fibroblasts in cluster 1 and CD55+ fibroblasts (SC-F4) in cluster 2. This is consistent with the recent cell-type abundance phenotype (CTAP) description by ref. 22, where the original pathotypes we described[18,24] were re-defined according to the different distribution of single cell subsets and validated in the R4RA cohort[19].

Of critical importance, the application of machine learning to the STRAP synovial gene expression at baseline identified predictive models of response at 16 weeks to each of the three drugs used in the trial (Fig. 5c, d). Moreover, in order to develop a predictive test of clinical utility in rheumatology practice, we converted the synovial RNA-Seq signatures into a nanoString panel of 524 genes and validated

it using residual synovial RNA from patients (*n* = 118) originally recruited to the STRAP trial. This analysis showed that each model was successfully validated with accuracy ranging from 79 to 85% and AUC ranging from 0.82 to 0.87 (Fig. 6b, c). Notably, prediction models for tocilizumab and rituximab were independently validated in an external cohort, namely the R4RA biopsy-driven randomised controlled trial (*n* = 133) (Fig. 5e)[19]. This validation is of critical importance for the generalisability of the predictive signatures since patients were recruited to the STRAP trial at an earlier disease stage, having only failed methotrexate-based DMARDs, while patients enroled in R4RA had failed both methotrexate-based DMARDs as well as at least one anti-TNF biologic and considered to be a more difficult to treat population[17]. These results indicate that synovial signatures reflect the diverse pathology linked to different pathways, truly representing the target of the different modes of action of these commonly used targeted biologic therapies.

A major achievement of this study has been the development of a novel algorithm which can assign patients to one of three classes of biologics (TNF-inhibitor, IL6-inhibitor, B-cell depleting agent) based on their probability of response predicted by machine learning models applied to nCounter assay on a patient's synovial biopsy (Fig. 6d). This algorithm can also predict whether patients have a low probability (all *p* < 0.5) of responding to all three drugs, in which case they are labelled as 'biomarker negative' and can be offered an alternative therapeutic class, thus reducing costs and unnecessary drug exposure to agents unlikely to be effective. This approach paves the way to precision prescribing in the future, contrary to the current inability to predict which drug the patient is likely to respond to, because of the lack, up until now, of predictive response biomarkers. While there has been a sustained research effort for decades to identify predictive peripheral blood biomarkers, none have been identified, as documented by multiple studies and meta-analyses[42]. Multiple studies have also investigated whether anti-CCP or rheumatoid factor titre or seropositivity predict response to biologics[43]. Some of the most comprehensive studies include a meta-analysis of four key RCT, which included 1416 rituximab-treated patients[44], and an observational study of 27,583 RA patients treated with four different biologics[45]. These show that seropositive patients have a tendency to modestly higher response rates when treated with rituximab or tocilizumab, but not anti-TNF inhibitors. Collectively, these studies reveal that the difference in response rates between seropositive and seronegative patients is modest and that anti-CCP and/or rheumatoid factor positivity are weak predictors of response. This is consistent with our own analysis of STRAP and R4RA which shows that CCP and RF titre as a continuous variable was not a strong predictor of response (Supplementary Fig. 8). However, RF and CCP positivity showed a modest association with increased response to rituximab and etanercept in STRAP (Supplementary Table 8), but the association with rituximab response was no

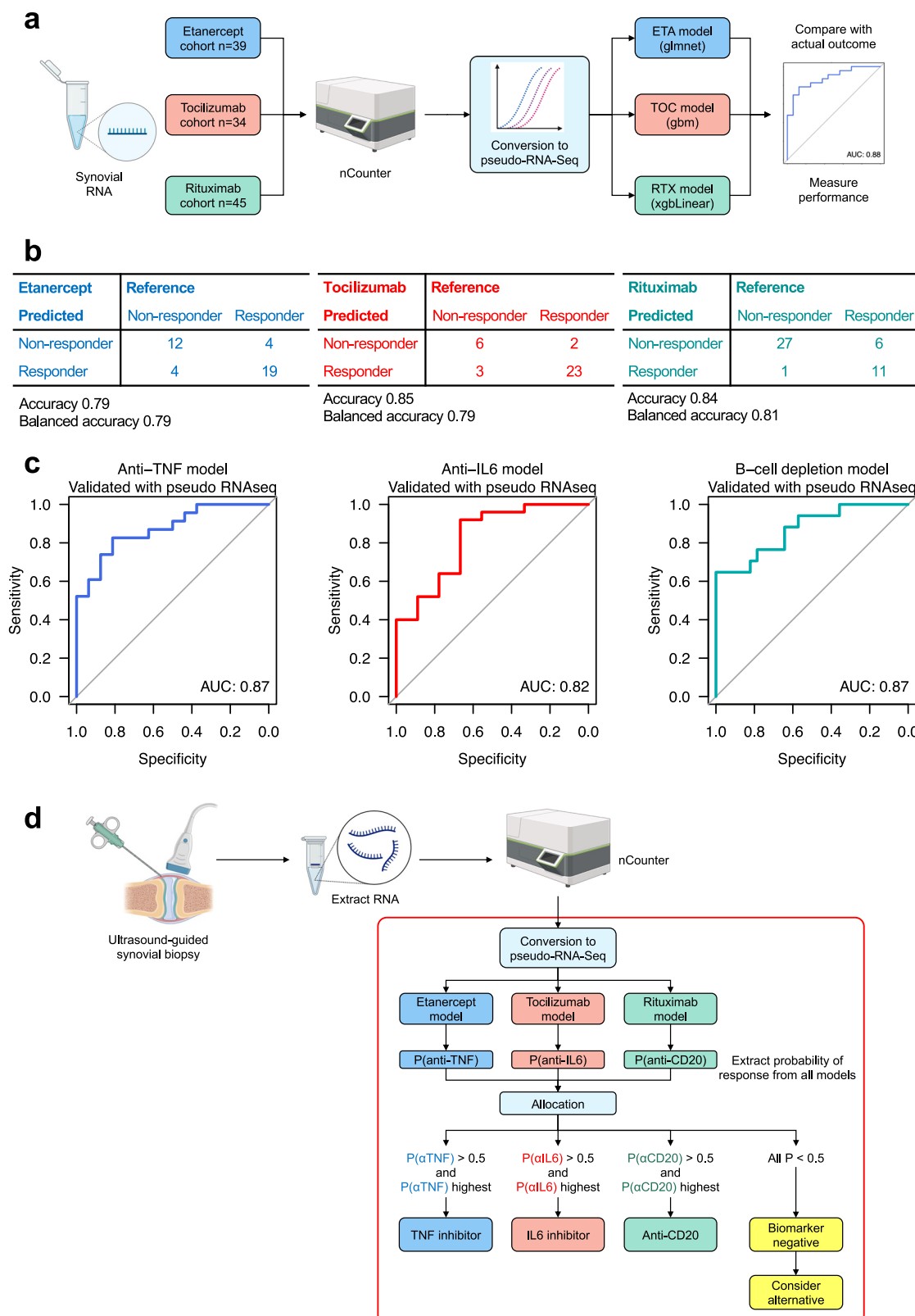

longer significant when STRAP and R4RA were combined. No significant association was observed for tocilizumab in STRAP/R4RA.

An important limitation of the present study, however, is that the precision, accuracy and clinical utility of the algorithm will require additional testing in a purposefully designed prospective RCT, which is currently ongoing, funded by the EU-Horizon-IMI programme: 3TR PRECIS-THE-RA[46]. The 3TR PRECIS-THE-RA trial will directly test

whether assigning RA patients to either a TNF-inhibitor or an IL6-inhibitor based on the probability of response determined through a synovial biopsy nCounter assay will enrich for clinical response compared to the control arm, in which patients will receive one of the two drugs randomly. Other important limitations of our study are that it is a post-hoc analysis, and there is a lack of replication for etanercept (no such biopsy RCT cohort exists). The CDAI score is preferred for

**Fig. 6 | Conversion and validation of machine learning models using the nCounter assay. a** Flow diagram outlining the process of converting the RNA-Seq models to a workable nanostring nCounter-based assay. Spare baseline synovial biopsy samples from STRAP were subjected to nCounter assay using a custom synovial 524-gene panel. nCounter data was rescaled to RNA-Seq scale ("pseudo-RNA-Seq") using linear models for each gene. Rescaled nCounter data was passed to machine learning models from Fig. 4c, and the performance of each model was assessed. **b** Confusion matrices showing predicted versus actual response, accuracy and balanced accuracy of nCounter assay applied to baseline synovial biopsies for prediction of response defined as DAS28-ESR <3.2 after 16 weeks of treatment.

**c** Receiver operating characteristic (ROC) curve plots and area under the curve (AUC) measurements for prediction of response to etanercept, tocilizumab and rituximab from nCounter assay applied to baseline synovial biopsies from STRAP. **d** Proposed algorithm for allocation of a new patient to one of three possible biologic therapy categories (TNF-inhibitor, IL6-inhibitor or B-cell depleting agent) based on whichever model gives the highest predicted probability of response. Individuals with low predicted probability (all $p < 0.5$) of response to all three classes of biologics are categorised as "biomarker negative" and can be offered an alternative class of therapeutic agent. **a, d** created in BioRender with modifications (https://BioRender.com/r4uqilh).

comparing anti-IL6 therapeutics against other drugs, such as placebo or standard of care, due to the absence of acute-phase reactants such as CRP, which are driven down by its mechanism of action and thus lead to artificially higher response rates according to measures such as DAS28-CRP[47]. But in our case, we are comparing tocilizumab responders vs non-responders, i.e. we are comparing tocilizumab against itself, so the tocilizumab non-responders still represent a valid non-response group which has less response to tocilizumab compared to the tocilizumab responder group. We attempted to fit models based on CDAI 50% response, but were unable to develop reliable models to predict this outcome measure for any of the three trial drugs. Our interpretation is that CDAI, which relies on both patient and clinician-reported measures and includes no biological measures, is more subjective. However, we were able to develop models which could predict response defined as DAS28-ESR <3.2. The fact that the DAS28 is less subjective in that it does not include a clinician-reported measure, plus the inclusion of ESR as an objective biological measure of systemic inflammation in the endpoint, might have aided the prediction modelling. The optimal choice of clinical response metrics is controversial and may vary depending on the drug being studied and the scientific question being asked—no response measure is perfect for all drugs and all clinical scenarios. Composite scores and subcomponents show major differences in correlation with functional outcomes such as future joint damage and function/disability[48]. Studies have shown that acute-phase reactants and swollen joint counts are the dominant predictors of radiographic joint damage[49,50]. Thus, the development of models which predict endpoints which include acute-phase reactants may have a long-term advantage for patients. Overall, however, we fully accept that in an ideal world, it would be preferable to predict CDAI response, especially for anti-IL6 therapeutics, but this might require substantially larger cohorts due to the higher subjectivity of the CDAI measure.

Sample size also affects the current study insofar as it can impact the reliability of model performance metrics such as AUC. Thus, we have provided full confusion matrices for all models (Supplementary Tables 5–7) as well as estimates of the variability of AUC and additional performance metrics across 25 repeats of the nested CV to aid interpretability.

Another limitation to mention is that current deconvolution algorithms are limited in accuracy for quantifying single-cell subsets from bulk RNA-Seq, and the abundance of single-cell subsets shown are only estimates. On the other hand, currently, it is not experimentally feasible to perform scRNA-Seq on such a large number of biopsies in a prospective RCT.

In summary, the in-depth analysis presented here has shown that machine learning identifies synovial biopsy biomarkers that predict response to three specific commonly used biologic classes, namely B-cell depleting agents, TNF- and IL6-inhibitors, targeting three major disease pathways. The machine learning biomarker models were successfully translated into an nCounter assay, which was validated in residual synovial RNA available from STRAP patients, with its performance tested against the actual response outcomes observed in the trial. Thus, this study provides the underpinning evidence towards changing the prescribing paradigm away from 'trial-and-error' to transform future clinical practice through successful bedside tests to help stratify patients to the most effective drugs from the disease outset. It also paves the way for patient-centric molecular-pathology driven clinical trials to develop new drugs for refractory patients[51].

## Methods

### Patients and intervention

A total of 226 patients aged 18 years or over, fulfilling 2010 ACR/EULAR classification criteria for RA who were eligible for treatment with anti-TNF therapy according to UK National Institute for Health and Care Excellence (NICE) guidelines, i.e. failing or intolerant to conventional synthetic disease-modifying anti-rheumatic drug (csDMARD) therapy were recruited when fulfilling the trial inclusion/exclusion criteria (for the full study protocol and baseline patient characteristics see the primary analysis of the trial)[20]. Due to issues with drug supply, a separate trial 'STRAP-EU' was opened which replicated the STRAP trial in the UK, and recruitment was expanded to four other countries in Europe in 2018. For the analyses, the data from both trials were combined. Briefly, patients underwent a synovial biopsy of a clinically active joint at entry to the trial, performed according to the expertise of the local centre as either an ultrasound-guided or arthroscopic procedure[52]. Following synovial biopsy, patients were randomised to receive either rituximab as two 1000-mg infusions at an interval of 2 weeks, administered at baseline, tocilizumab 162 mg administered as a weekly subcutaneous injection, or etanercept 50 mg administered as a weekly subcutaneous injection. Patients were followed up every 4 weeks (±1 week) throughout the 48-week trial treatment period, where RA disease activity measurements and safety data were collected. Follow-up of STRAP patients recruited in the UK after 1st January 2019 ended at 24 weeks from baseline, whereas the follow-up of all STRAP-EU patients continued to end at 48 weeks from baseline. Optionally, a repeated synovial biopsy of the same joint sampled at baseline was performed at 16 weeks. The study was conducted in compliance with the Declaration of Helsinki, International Conference on Harmonisation Guidelines for Good Clinical Practice, and local country regulations. The final protocol, amendments and documentation of consent were approved by the institutional review board of each study centre and relevant independent ethics committees: UK ethics committee approval MREC 14/WA/1209 (Wales REC 3); Comité d'Ethique Hospitalo-Facultaire Saint-Luc, Bruxelles, Belgium; Comitato Etico Interaziendale, A.O.U. Maggiore della Carita, Novara, Italy; Comissão de Ética para a Investigação Clínica (CEIC), Portugal; Comité Ético de Investigación con medicamentos (CEIm) del Hospital Clínic de Barcelona, Spain. All patients provided written informed consent. The trial was supported by an unrestricted grant from the Medical Research Council. The study protocol is available at http://www.matura-mrc.whri.qmul.ac.uk/documents.php. These trials are registered with the EU Clinical Trials Register, 2014-003529-16 (STRAP) and 2017-004079-30 (STRAP-EU).

### RNA-seq data processing and analysis

A total of 283 paired-end RNA-seq samples from 50 million reads of 150-bp length were mapped to the reference human transcriptome (Gencode v29, GRCh38.p12), and transcripts were then quantified

using Salmon version 0.13.1[53]. Tximport version 1.13.10 was used to aggregate transcript-level expression data to genes, then counts were subjected to variance-stabilising transformation (VST) using the DESeq2 version 1.25.9 package[54]. Following RNA-Seq quality control, with principal component analysis (PCA), five baseline and three follow-up, in total eight samples were excluded due to poor mapping rate that was originated from low RNA quality (Supplementary Fig. 1a). Thus RNA-Seq data from 208 patients were available for subsequent analysis at baseline (65 samples at later time points are not analysed here). Baseline characteristics of patients with available RNA-Seq are shown in Supplementary Table 1.

### Histological analysis

A minimum of six synovial biopsies were processed in an Excelsior tissue processor before being paraffin-embedded en masse at Queen Mary University of London Core Pathology department. Tissue sections (3–5-μm thickness) were stained with hematoxylin and eosin and IHC markers CD20 (B cells), CD138 (plasma cells), CD21 (follicular dendritic cells) and CD68 (macrophages) in an automated Ventana Autostainer machine. CD79A (B cells) and CD3 (T-cells) staining was performed in-house on deparaffinized tissue following antigen retrieval (30 min at 95 °C), followed by peroxidase- and protein-blocking steps. Primary antibodies (CD79A (clone JCB117, Dako), CD3 (clone F7.238, Dako), CD20 (clone L26, Dako), CD68 (clone KP1, Dako) and CD138 (clone MI15, Dako)) were used for 60 min at room temperature. Visualisation of antibody binding was achieved by 30-min incubation with Dako EnVisionTM+ before completion by the addition of 3,3′-diaminobenzidine (DAB) + substrate chromogen for 10 s, followed by counterstaining with hematoxylin. Following IHC staining, sections underwent semiquantitative scoring (0–4), by a minimum of two assessors, to determine levels of CD20+ and CD79a+ B cells, CD3 + T-cells, CD138+ plasma cells and CD68+ lining (L) and sublining (SL) macrophages, adapted from a previously described score[55] and subsequently validated[18]. Hematoxylin and eosin-stained slides also underwent evaluation to determine the level of synovitis according to the Krenn synovitis score (0–9)[56]. Synovial biopsies were classified into synovial histological patterns, also known as pathotypes, according to the following criteria: (1) lymphomyeloid, presence of grade 2–3 CD20+ aggregates, CD20 ≥2 and/or CD138 ≥2; (2) diffuse-myeloid, CD68SL ≥2, CD20 ≤1 and/or CD3 ≥1 and CD138 ≤2; and (3) pauci-immune-fibroid, CD68SL <2 and CD3, CD20 and CD138 <1.

### Differential expression, modular and pathway analysis of RNA-Seq data at baseline

Patients classed as responders and non-responders based on their 16-week assessment using ACR20 criteria were compared for each individual treatment group as well as for all treatments combined. The groups showed no significant differences for baseline characteristics, including histological and molecular B-cell status, gender or disease duration (Supplementary Table 1). Low-expressed genes (expressed in fewer than 18 samples with at least a normalised count of 9) were excluded from analysis. Remaining genes were subjected to differential gene expression analysis based on general linear regression models with negative binomial distribution applied to RNA-Seq count data using DESeq2 (version 1.34.0), which uses a Wald test to compare differences between treatment response groups in synovium RNA-Seq samples. To prevent bias of results through potential muscle contamination, a supervised principal component analysis (PCA) was calculated using the `prcomp` function with the following 17 genes derived from Reactome skeletal muscle module: *ACTA1, ACTN2, MYBPC1, MYBPC2, MYH1, MYH2, MYH7, MYH8, MYL1, MYL2, NEB, TCAP, TNNC2, TNNI1, TNNI2, TNNT1* and *TNNT3*. These genes were checked and confirmed to be highly specific to muscle tissue in the FANTOM5 CAGE-Seq repository. Muscle gene-specific PC1 was employed as a covariate in the DESeq2 analysis to adjust for the presence of small

amounts of muscle tissue in a few samples. *P* values were false discovery rate (FDR) adjusted using Storey's *q* value. A cut-off of *q* < 0.05 was used to identify significantly differentially expressed genes (DEG), illustrated by volcano plots. DEG analysis was repeated with adjustment for biopsy joint size (large = knee, ankle, elbow; small = MCP, PIP, MTP, wrist), but this had negligible impact on DEG analysis and only worsened the number of DEG identified for two of the drugs (Supplementary Fig. 2). Biopsy joint size was not significantly different between responder and non-responder groups (Supplementary Table 2).

DESeq2 outputs from each individual treatment group were used for modular analysis using the Bioconductor package quantitative set analysis for gene expression (QuSAGE, v2.30.0). Weighted gene correlation network analysis (WGCNA) gene modules from ref. 57 were selected for gene set enrichment, and relevant modules were summarised in plots. DESeq2 outputs from the differential expression analysis in all treatments were used for functional enrichment using enrichR (v3.2) with *Reactome_2022* as the pathway repository.

### 2 × 3-way analysis of differentially expressed genes

In order to visualise the specificity of genes for predicting responsiveness to each drug, DEGs identified by the responders vs non-responders contrast in each drug cohort were displayed in three-way polar plots using the volcano3D R package (version 2.0.6) as a 2 × 3-way analysis[18]. DEGs upregulated in responders were categorised as E+, R+, T+ based on the Wald Chi-squared test *p* value significance for responder vs non-responder analysis for etanercept, rituximab and tocilizumab, respectively. Genes whose FDR result was significant for two drugs were labelled as mixed categories (E+ R+, E+ T+ or R+ T+). Genes significantly upregulated in responders with all three drugs were excluded. The graphical position of the genes along the three axes of the polar plot was calculated using the estimated log fold change for each gene. The same procedure was applied with directionality reversed to plot genes significantly upregulated in the non-responder group of each drug.

### Gene expression integration into cell-specific modules

Modular approaches for gene set enrichment analysis and relative quantification of cell subsets are widely used in recent molecular studies[58,59]. Here, we integrated gene expression into cell-specific modules to characterise the association of synovial immune cells in RA with multidrug resistance. For the enrichment of 18 single-cell subsets identified in scRNA-Seq of RA synovial tissue[21] that are composed of four fibroblast subtypes (SC-F1: CD34+ sublining, SC-F2: HLA+ sublining, SC-F3: DKK3+ sublining and SC-F4: CD55+ lining), four macrophages subtypes (SC-M1: IL1B+ pro-inflammatory, SC-M2: NUPR1+, SC-M3: C1QA+, SC-M4: IFN-activated), six T-cell subtypes (SC-T1: CCR7+ CD4+, SC-T2: FOXP3+ Tregs, SC-T3: PD-1+ Tph/Tfh, SC-T4: GZMK+ CD8+, SC-T5: GNLY+ GZMB+, SC-T6: GZMK+/GZMB+), and four B-cell subtypes (SC-B1: IGHD + CD27- naive, SC-B2: IGHG3+ CD27+ memory, SC-B3: Autoimmune associated, SC-B4: Plasmablasts), we scored each subtype using a modular approach that integrates previously published gene signatures[21]. The top five exclusively differentially expressed genes (based on AUC scores) were utilised as cell subtype-specific gene sets. Module scores for each subtype were calculated using the `AddModuleScore` function from the R package Seurat. Linear model method as implemented in R package Limma, was used to detect the differentially abundant cell sub-populations when comparing responders and non-responders.

The main aim of the STRAP clinical trial is to test the utility of analysing synovial B-cell infiltrates as a potential biomarker to guide therapeutic decisions in patients failing DMARD therapy. To stratify patients according to their synovial B-cell infiltrates into B-cell poor/rich pathotypes, we used the B-cell module gene set that contains 71 genes derived from FANTOM5 (see Supplementary Fig. 1c)[60]. First, we

assigned the mean of normalised and scaled gene expression values to this module as previously described and validated[18]. Then, patients were categorised as B-cell poor or B-cell rich using a predefined cut-off of −0.0413, the median B-cell module values of RA patients recruited in the R4RA clinical trial[17].

### Unsupervised clustering of baseline gene expressions

To perform unbiased heatmap clustering on the entire baseline population, genes were filtered based on expression levels. For the STRAP cohort, genes were stringently filtered to only highly expressed genes using the edgeR package (version v4.2.0) function `filter-ByExpr` with arguments `min.count` set to 20 and `min.total.count` set to 5e5. Since R4RA had a smaller sample size ($n = 133$), `min.total.count` was reduced to 3e5. Expression levels underwent variance-stabilising transformation (VST) normalisation and Z-score scaling for visualisation with ComplexHeatmap (v2.14.0). Euclidean distance was employed to calculate row distances, while Pearson correlation distance was utilised for column distances. Hierarchical clustering was performed on columns using the complete linkage method, whereas k-means clustering was applied to rows. Genes within each resulting cluster were used for pathway analysis with enrichR (v3.2) using the *Reactome_2022* database.

### Building classifier models for the prediction of response

Machine learning models were built to predict target DAS28$_{ESR}$, target DAS28$_{CRP}$, ACR20 response, CDAI 50% response, EULAR DAS28-ESR good vs moderate/non-response and EULAR DAS28-CRP good vs moderate/non-response to either rituximab, tocilizumab, or etanercept treatment at the primary endpoint (16 weeks).

The model feature space was created using RNA-Seq data restricted to 507 target genes relevant to synovial biology based on the nCounter custom panel (see description of the panel below). Baseline clinical parameters were included to improve response prediction. These included: tender joint count (TJC), swollen joint count (SJC), arthritis activity (patient visual analogue score), rheumatoid factor titre (RF), anti-CCP titre, erythrocyte sedimentation rate (ESR) and C-reactive protein (CRP). TJC, SJC and ESR were square root transformed (sqTJC, sqJSC and sqESR) to be on a Gaussian distribution, and CRP was log-transformed (logCRP). Genes were filtered to those with mean expression ≥6 on the VST scale to remove low-expressed genes, which were less reliably detected by nCounter.

Following processing, data was split into $10 \times 10$ nested CV folds using the nestedcv R package (version 0.7.9)[31]. Feature selection was performed within outer CV folds using a *t*-test filter, with the top *n* genes by two-tailed *t*-test *p* value being retained for fitting of models. An alternative feature selection method, which was used for rituximab response prediction models, was based on fitting a glmnet elastic net regression model to the whole data and selecting genes which were retained in the glmnet model. The number of features selected was chosen to limit model size to between 25 and 40 predictors to design practical models more likely to be feasible in real-life clinical situations. Model hyperparameters were tuned by an inner tenfold cross-validation based on log loss. Overall model performance was determined by tenfold outer cross-validation with 25 repeats to give averaged unbiased estimates of model accuracy. Elastic net penalised regression using the glmnet package was compared against seven machine learning models from the caret package (version 6.0): random forest (RF), least-squares support vector machine (SVM) with radial basis function kernel (svmRadial), least-squares SVM with polynomial kernel (svmPoly), gradient boosted machine (GBM), mixture discriminant analysis (MDA), extreme gradient boosting (xgboost) using trees (xgbTree) or linear regression (xgbLinear). None of these models, with the exception of glmnet, are sparse, which means that they incorporate all predictors during modelling. This leads to models with large numbers of predictors, which have a tendency to fail

validation in real clinical situations. Hence, feature selection was used to limit model size to between 25 to 40 predictors to design practical models more likely to be feasible in a real-life clinical situation. Previous studies have shown that with gene expression data, filtering genes with a simple *t*-test often performs better than more complex feature selection methods[61]. Thus, for models predicting etanercept or tocilizumab response, we used a *t*-test filter. However, it was more difficult to obtain a good-quality prediction model for rituximab, so a glmnet-based filter was used. In this two-step process, a LASSO regression model was fitted to the training folds to select optimal features, which are then passed to other models for fitting.

To evaluate overall model performance from each repeat of $10 \times 10$-fold nested CV, predictions from left-out outer CV test folds were pooled, and performance was determined compared to the ground truth. Multiple metrics were used including area under curve (AUC) from receiver operating characteristic (ROC) curves computed using R package pROC, accuracy and balanced accuracy. Tuning parameters for the final model were determined by a final round of CV on the whole dataset with the final model fitted to the whole dataset.

Performance of each model for predicting response to each drug was measured by AUC (area under ROC curve) with 25 repeats of $10 \times 10$ nested CV to compare the primary endpoint ACR20 response against five other response endpoints to identify the optimal endpoint (Supplementary Figs. 5–7): DAS28-ESR/CRP <3.2 (target DAS28-ESR/CRP); CDAI 50% response, EULAR good vs moderate/non-response for DAS28-ESR/CRP. This showed that for etanercept and tocilizumab, the best prediction of response was seen for target DAS28-ESR (<3.2). Therefore, this response outcome measure was selected for model optimisation. For etanercept and tocilizumab, models were trained with a binary outcome, with response defined as DAS28-ESR <3.2 at 16 weeks. For the rituximab model, due to difficulties with obtaining a reliable binary classification model, an ordinal outcome was used, namely DAS28-ESR status at 16 weeks which has four levels: high (DAS28-ESR >5.1), moderate (3.2 < DAS28 <5.1) or low disease activity (2.6< DAS28 <3.2) and remission (DAS28 <2.6). This four-level outcome fits alongside the original binary outcome as low disease activity/remission corresponds directly to response, and moderate/high disease activity at 16 weeks corresponds to non-response. Rituximab prediction models were fitted to this ordinal outcome as a regression, then converted to a binary outcome after the final model was fitted and performance calculated for the binary outcome.

Feature importance was measured across outer CV folds as well as the final model to rank predictors in terms of importance and estimate stability of variables in models by the frequency with which variables were selected across outer CV folds and estimate the variance across variable importance across the outer CV folds and final model.

### Validation of machine learning models in R4RA

Two of the machine learning models built on STRAP data were validated in the independent R4RA cohort. In the R4RA trial, patients ($n = 164$) were randomised to tocilizumab or rituximab (not etanercept or anti-TNF)[17]. Therefore, only these two models were tested in R4RA. Of the 164 patients randomised on entry to the R4RA trial, good-quality RNA-Seq was available on $n = 133$ post-QC[19]. R4RA baseline synovial RNA-Seq data ($n = 133$) was batch corrected against STRAP RNA-Seq data using the `ComBat` function from the SVA R/Bioconductor package (version 3.52.0). Baseline clinical parameters required for each predictive model underwent identical transformation as in STRAP (square root for TJC, SJC, ESR; log for CRP). The tocilizumab and rituximab model-predicted response was compared against the actual DAS28-ESR response defined as DAS28-ESR <3.2 in individuals treated with tocilizumab ($n = 65$) or rituximab ($n = 68$) at the 16-week primary endpoint of the R4RA trial. Model performance for predicting response to each drug was measured by ROC AUC.

## Development of the custom synovium nCounter panel

The nCounter platform provides the flexibility to customise assay content with up to 800 user-defined targets to meet specific project demands. Our custom panel, also called the biologic screening panel for arthritis, was created using an iterative development process. The initial panel was designed with five gene sets that contained a total of 798 unique genes related to synovial pathobiology. These gene sets included: (i) genes associated with synovial pathotypes from the PEAC RNA-Seq study[18] (258 genes), (ii) protein-protein interaction networks of RA drug targets retrieved from the STRING database (https://www.string-db.org, 166 genes), (iii) gene predictors from previously developed treatment response prediction models (320 genes)[19,24,30], cell-type-specific genes from single-cell and spatial transcriptomics datasets (147 genes)[21,22,33,62,63], and 20 housekeeping genes for data normalisation identified through analysis of PEAC, R4RA and STRAP RNA-Seq data. Even though nCounter panels are completely customisable, the content of the manufactured panels may differ significantly from the original design because not all genes have probes available. In our case, issues with the probe design led to the exclusion of 33 genes from the panel before the manufacturing of the first cartridge. Then we tested our first panel on 48 samples, including synovium RNA isolates, Universal Human Reference RNA (Thermo Fisher QS0639) and water. Unexpectedly, in all samples, the negative control probes generated signals which were excessively strong for QC, with negative signals apparently higher than signals collected from some valid gene probes. We observed that from the first iteration that, in order to prevent extreme outliers from suppressing the signal of the other probes and reduce the noise signal in synovial tissue, it was optimal for application to synovial tissue to reduce the content of custom panels to around 600 genes with probes within a specific dynamic range of expression levels. Therefore, to refine the gene list, less informative genes which were either excessively highly expressed (an average count of >10,000) or very low expressed (maximum counts <100) were removed from the panel. Selected problematic probes were redesigned by Nanostring scientists, when possible. Probes identified as problematic ('sticky') by the manufacturing company were removed, as well as probes located in untranslated regions.

In the second round, we submitted a refined list of 458 genes with an additional gene set that was a collection of genes associated with JAK-STAT, TNF and IFN signalling pathways identified from the KEGG database (206 genes). This resulted in 526 unique genes. Three gene probes were subsequently discarded from the panel as their probes had low specificity and were predicted to predispose to high background signal. The second version was synthesised with 523 unique genes and tested on a further 48 synovial samples. Subsequently, the *LYZ* gene probe was redesigned and included in the final, third version of the panel. Test runs of version 2 onwards did not report abnormal signals from control probes, and the final version with 524 genes (507 target genes and 17 housekeeping genes, Supplementary Data 5) was manufactured for this study. Expression stability of the housekeeping genes was assessed using data collected from the test runs, and due to less stable expression patterns, three housekeeping genes (*ISY1*, *STK11IP* and *TFRC*) were removed from the data normalisation process. Supplementary Fig. 4 illustrates the entire process and contains all the information regarding the development of the custom panel.

## nCounter analysis

The final custom synovium nanoString nCounter panel developed contained 507 genes relevant to synovial pathobiology based on previous studies of synovial gene expression (Supplementary Data 5)[18,19,24,30]. The panel also contained 17 housekeeping genes (i.e. a total of 524 genes) selected for detectable and stable expression in synovial tissue and technical probes (six positive and eight negative). RNA samples were assayed using the nanoString nCounter Sprint Profiler from 100 ng of synovial tissue RNA, following the manufacturer's instructions. Raw nCounter counts were extracted from RCC files and normalised using nanoString's official R package (nanoStringNCTools version 1.6.0). Housekeeping genes and panel standard probes (synthetic oligos) normalisation methods were applied to the raw data to scale and standardise the data.

## nCounter pseudo-RNA-Seq conversion and prediction using machine learning models

Linear models for each gene were fitted between variance stabilised transformed bulk RNA-Seq gene counts and log-transformed normalised nCounter gene counts. The nCounter assay data were converted to RNA-Seq scale (referred to as pseudo-RNA-Seq) based on stored regression models for each gene. Using predefined linear regression components (intercept and slope coefficient), nCounter data was converted to pseudo-RNA-Seq and then passed as fresh data input into the finalised machine learning fitted nestedcv package model objects (glmnet model for etanercept; gbm model for tocilizumab; xgbLinear model for rituximab) to determine a predicted probability of response for each sample. Predicted response probability was compared with actual outcome in the STRAP trial to determine confusion matrices of predicted binary response vs actual response, accuracy, balanced accuracy and ROC AUC for the nCounter assay.

## Statistical analysis

Statistical analysis of RNA-Seq gene expression data were performed using negative binomial general linear models using the DESeq2 R package. For differential gene expression, *p*-values were false discovery rate (FDR) adjusted using Storey's *q* value. A cut-off of $q < 0.05$ was considered significant. Statistical analysis of gene modules was performed using QuSAGE with FDR adjustment, and a cut-off of $q < 0.05$ was considered significant (unless otherwise stated). Two-tailed tests were used throughout.

## Reporting summary

Further information on research design is available in the Nature Portfolio Reporting Summary linked to this article.

# Data availability

The datasets generated during and/or analysed during the current study are available on an interactive web interface that allows direct data exploration (https://strap.hpc.qmul.ac.uk/). A searchable interface allows users to examine relationships between individual synovial gene transcript levels and histological and clinical parameters, and clinical response at 16 weeks. The website was constructed using R Shiny server 1.5.16, with interactive plots generated with R plotly 4.9.3. RNA-Seq data is available at ArrayExpress accession ID E-MTAB-13733. All data are included in the Supplementary Information or available from the authors, as are unique reagents used in this Article. The raw numbers for charts and graphs are available in the Source Data file whenever possible. Source data are provided with this paper.

# Code availability

The nestedcv R package[31] used to build and test the machine learning models is publicly available for installation from the CRAN R repository (https://doi.org/10.32614/CRAN.package.nestedcv). The source code is also available on GitHub at https://github.com/myles-lewis/nestedcv. Scripts used for figure generation, model building and performance testing are available from https://github.com/EMR-bioinformatics/STRAP.

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

## Acknowledgements

The STRAP trial was part of the Maximising Therapeutic Utility in RA (MATURA) programme, jointly funded by the UK Medical Research Council (MRC) and Versus Arthritis (grant reference MR/K015346/1). This study also received support from the UK National Institute for Health Research (NIHR) grant 131575, MRC grant MR/V012509/1 and European Commission Innovative Medicines Initiative grant 831434 (3TR). Infrastructure support was provided by Versus Arthritis (Experimental Arthritis Treatment Centre grant, number 20022). We thank the Patient Advisory Group, represented by Zoe Ide (Chair), for their continuous support in evaluating trial documentation and linked research as well as in disseminating the importance of precision medicine among patients and through patient organisations, including the National Rheumatoid Arthritis Society. This study was supported by NIHR Manchester, Leeds, Newcastle, Birmingham, Cambridge, Oxford, Southampton, Barts and University College London Hospital Biomedical Research Centres and NIHR Clinical Research Facilities; the views expressed are those of the authors and not necessarily the UK National Health Service, NIHR, or the Department of Health. We thank all patients participating in the trial for consenting to a minimally invasive procedure that they would not have had as part of routine care.

## Author contributions

C.P. conceived the trial, sought funding and assumed overall responsibility for the clinical trial. M.J.L., C.C., A.E.A.S., E.S., R.L., K.G. and G.G. performed data analysis. L.F.-J. was responsible for the collection and processing of samples. A.N., F.R., M.J.L. and C.P. oversaw the clinical trial. G.G. performed statistical analysis of the clinical trial. M.J.L. and C.C. performed the machine learning and algorithm development. M.J.L., K.G. and C.C. created the shiny website. M.J.L. and C.P. supervised the data analysis and took overall responsibility for the data. The first draft of the manuscript was written by M.J.L., C.C., A.E.A.S., E.S. and revised by M.J.L. and C.P. All authors contributed to the discussion and interpretation of the results, critically reviewed the manuscript and approved the final version for submission.

## Competing interests

C.P., M.J.L. and C.C. are inventors on a patent application (no. GB 2410224.6), submitted by Queen Mary University of London, that covers methods used to select treatments in RA. The remaining authors declare no competing interests.

## Additional information

## the STRAP collaborative group

Felice Rivellese [1,2], Alessandra Nerviani [1,2], Giovanni Giorli[1], Louise Warren[1], Edyta Jaworska[1], Michele Bombardieri[1,2], Myles J. Lewis [1,2,37] ✉, Frances Humby[5], Arthur G. Pratt[6], Andrew Filer[7], Nagui Gendi[8], Alberto Cauli[9], Ernest Choy[10], Iain McInnes[11], Patrick Durez[12], Christopher J. Edwards[13], Maya H. Buch[14], Elisa Gremese[15], Peter C. Taylor[16], Nora Ng[5], Juan D. Cañete[17], Sabrina Raizada[18], Neil D. McKay[19], Deepak Jadon[20], Pier Paolo Sainaghi[21], Richard Stratton[22], Michael R. Ehrenstein[23], Pauline Ho[24], Joaquim P. Pereira[25], Bhaskar Dasgupta[26], Claire Gorman[27], Ahmed Zayat[28], Ana Rita Machado[25], Andrea Cuervo[29], Arti Mahto[30], Charlotte Rawlings[10], Chijioke Mosanya[6], Christopher D. Buckley[31], Chris Holroyd[13], Deborah Maskall[14], Francesco Carlucci[31], Georgina Thorburn[1], Gina Tan[1], Gloria Lliso-Ribera[1], Hasan Rizvi[2], Joanna Peel[1], João Eurico Fonseca[25], John D. Isaacs[6], Julio Ramírez[32], Laurent Meric de Bellefon[12], Mary Githinji[1], Mattia Congia[9], Neal Millar[11], Nirupam Purkayastha[1], Rakhi Seth[13], Raquel Celis[32], Rebecca Hands-Greenwood[1], Robert Landewé[33], Simone Perniola[15], Stefano Alivernini[15], Stefano Marcia[34], Stefano Marini[34], Stephen Kelly[2], Vasco Romão[25], James Galloway[30], Hector Chinoy[14], Désirée van der Heijde[35], Peter Sasieni[36], Anne Barton[14] & Costantino Pitzalis [1,2,4,37] ✉

[5]Rheumatology Department, Guy's and St Thomas' NHS Foundation Trust, London, UK. [6]Translational and Clinical Research Institute, Newcastle University, Newcastle upon Tyne, UK. [7]Rheumatology Research Group, Institute for Inflammation and Ageing, NIHR Birmingham Biomedical Research Centre, University of Birmingham, Queen Elizabeth Hospital, Birmingham, UK. [8]Basildon and Thurrock University NHS Hospitals Foundation Trust, Basildon, UK. [9]Rheumatology Unit, AOU and University of Cagliari, Monserrato, Italy. [10]CREATE Centre, Cardiff University, Cardiff, UK. [11]Glasgow Clinical Research Facility, Glasgow Royal Infirmary, Glasgow, UK. [12]Institute of Experimental and Clinical Research, Université Catholique de Louvain, Brussels, Belgium. [13]NIHR Southampton Clinical Research Facility, University Hospital Southampton, Southampton, UK. [14]Centre for Musculoskeletal Research, Division of Musculoskeletal and Dermatological Sciences, Faculty of Biology, Medicine and Health, University of Manchester, Manchester, UK. [15]Fondazione Policlinico Universitario Agostino Gemelli IRCCS, Rome, Italy. [16]Nuffield Department of Orthopaedics, Rheumatology and Musculoskeletal Sciences, Botnar Research Centre, University of Oxford, Oxford, UK. [17]Institut d'Investigacions Biomèdiques August Pí I Sunyer, Barcelona, Spain. [18]New Cross Hospital and Cannock Chase Hospital, Royal Wolverhampton NHS Trust, Wolverhampton, UK. [19]Edinburgh Rheumatology Research Group and Rheumatic Diseases Unit, NHS Lothian, Edinburgh, UK. [20]Department of Medicine, University of Cambridge, Cambridge, UK. [21]Department of Rheumatology, University of Eastern Piedmont and Maggiore della Carita Hospital, Novara, Italy. [22]Royal Free Hospital, Royal Free London NHS Foundation Trust, London, UK. [23]University College Hospital, University College London Hospitals NHS Foundation Trust, London, UK. [24]The Kellgren Centre for Rheumatology, Manchester Royal Infirmary, Manchester University NHS Foundation Trust, Manchester, UK. [25]Rheumatology Research Unit, Instituto de Medicina Molecular João Lobo Antunes, Faculdade de Medicina, Universidade de Lisboa, Lisbon, Portugal. [26]Rheumatology Department, Mid & South Essex University Hospitals NHS Foundation Trust, Southend University Hospital, Westcliff-on-Sea, UK. [27]Department of Rheumatology, Homerton University Hospital, Homerton Healthcare NHS Foundation Trust, London, UK. [28]Leeds Teaching Hospitals NHS Trust, NIHR Leeds Biomedical Research Centre, Leeds, UK. [29]Rheumatology Department, Hospital General de Granollers, Granollers, Spain. [30]King's College Hospital, King's College Hospital NHS Foundation Trust, London, UK. [31]Kennedy Institute of Rheumatology, University of Oxford, Oxford, UK. [32]Rheumatology Department, Hospital Clinic, Barcelona, Spain. [33]Department of Clinical Immunology & Rheumatology, Zuyderland Medical Centre, Sittard-Geleen, Heerlen, Netherlands. [34]UOC Radiologia SS, Trinità Hospital, Cagliari, Italy. [35]Department of Rheumatology, Leiden University Medical Center, Leiden, Netherlands. [36]Wolfson Institute of Population Health, Queen Mary University of London, London, UK.

