## [Peer Review file · Nature Communications]

Deep molecular profiling of synovial biopsies in the STRAP trial identifies signatures predictive of treatment response to biologic therapies in Rheumatoid Arthritis

Corresponding Author: Professor Myles Lewis

Version 0:

Reviewer comments:

Reviewer #1

(Remarks to the Author)

The manuscript presents a compelling and potentially impactful study exploring the prediction of treatment responses to biologic therapies in Rheumatoid Arthritis. Leveraging RNA-sequencing data from the STRAP trial alongside machine learning to develop predictive models is a notable strength of this work. Additionally, the validation in an independent cohort and the adaptation of findings into an nCounter panel for potential clinical application are highly valuable contributions. That said, addressing certain statistical and methodological aspects could further enhance the rigor, reproducibility, and overall clarity of the findings.

Availability of Code and Scripts

The analysis of RNA-Seq data and the development of machine learning models offer significant insights into predicting treatment responses in RA. However, the absence of publicly accessible code and scripts presents a challenge to independently verifying the results and fully understanding the methodology employed.

1. Supporting Reproducibility Standards

Reproducibility is a fundamental pillar of scientific research. Many journals and funding bodies increasingly advocate for or require code sharing to allow independent validation and build upon published work [Peng, R. D. (2011). *Science*, 334(6060), 1226-1227; Sandve, G. K., et al. (2013). *PLoS Computational Biology*, 9(10), e1003282]. The FAIR principles further underscore the value of making data and code findable, accessible, interoperable, and reusable to benefit the broader research community [Wilkinson, M. D., et al. (2016). *Scientific Data*, 3, 160018].

2. Enhancing Methodological Clarity

Providing analysis scripts would greatly clarify several key steps that remain somewhat ambiguous in the manuscript:

- RNA-Seq Data Analysis: Scripts could specify quality control metrics, trimming parameters, alignment tools (e.g., STAR, HISAT2) with versions, and the normalization approach in DESeq2 (e.g., relative log expression or trimmed mean of M-values).
- Batch Effect Management: Code would indicate whether batch effects (e.g., from sequencing runs or biopsy sites) were addressed using methods like ComBat or SVA, and how these adjustments were incorporated into DESeq2.
- Differential Gene Expression: Scripts would detail the DESeq2 model formula, including covariates (e.g., age, sex, disease duration) beyond biopsy joint size, as well as gene filtering and p-value adjustment criteria.
- Machine Learning Models: Code would elucidate feature selection (e.g., t-test or ANOVA thresholds, LASSO tuning), algorithm details (e.g., random forest hyperparameters), and the implementation of nested cross-validation and SMOTE oversampling.
- Independent Validation (R4RA): Scripts would confirm consistent processing between STRAP and R4RA cohorts,

including normalization and batch effect alignment.

3. Fostering Transparency and Confidence

Sharing scripts would enhance transparency, allowing readers to evaluate the appropriateness of the methods and identify any potential limitations, thereby strengthening trust in the study's conclusions.

Suggestion

To address this, I kindly recommend that the authors consider depositing all code and scripts used for data processing, statistical analysis, and model development in a public repository (e.g., GitHub, GitLab). The repository could include:

- Clearly documented scripts with step-by-step explanations.
- Instructions for replicating the analysis from raw data to final results.
- Software version details (e.g., R, Python, DESeq2, scikit-learn).

This step would align the study with contemporary standards for reproducible research and significantly bolster its credibility.

If sharing raw data is restricted due to privacy concerns, the authors might consider:

- Providing Markdown files or Jupyter notebooks with:
 - Complete analysis scripts.
 - Code to load intermediate data (e.g., normalized expression matrices, DESeq2 results, feature-selected datasets), shareable without raw data.
 - Guidance on running the scripts to replicate key findings from these intermediates.

This approach would balance transparency with patient confidentiality, enabling reviewers and researchers to follow the workflow without accessing sensitive data.

Statistical Observations

The manuscript reports AUC values of 0.786 (rituximab, n=65) and 0.713 (tocilizumab, n=68) for the R4RA validation cohort, which are promising results. However, the Abstract and Discussion sections could benefit from explicitly noting the modest sample sizes to provide context and avoid overstating the reliability of these estimates.

1. Impact of Small Validation Cohorts on AUC Stability

AUC values derived from small cohorts ($n < 100$) can exhibit considerable variability and may overestimate true performance due to limited data points shaping the ROC curve [Hanley, J. A., & McNeil, B. J. (1982). *Radiology*, 143(1), 29-36]. In smaller datasets, the ROC curve may appear "step-like" because fewer distinct prediction thresholds are available, potentially inflating the AUC [Lobo, J. M., et al. (2008). *Global Ecology and Biogeography*, 17(2), 145-151]. This effect stems from the discrete jumps in True Positive Rate (TPR) and False Positive Rate (FPR) with fewer samples, rather than the smooth curve typical of larger datasets, which can exaggerate class separation if overfitting or chance alignments occur.

In small datasets, the AUC might be inflated because the limited thresholds can exaggerate the separation between classes, especially if the model overfits or if random chance aligns the few data points favorably. The AUC is calculated as the integral under the ROC curve, and with fewer steps, the area can appear artificially high if the steps occur at points that maximize TPR while minimizing FPR, even if this doesn't reflect true generalizability. This inflation is particularly problematic because AUC is often interpreted as the probability that a randomly chosen positive instance is ranked higher than a negative one—a metric that assumes a robust sampling of the underlying distribution, which small datasets can't reliably provide.

- Fawcett (2006) [Pattern Recognition Letters] notes that ROC granularity depends on sample size, with fewer points creating a step-like curve.
- Hanley and McNeil (1982) emphasize AUC's sample-size dependency.
- Berrar and Flach (2012) [Briefings in Bioinformatics], Davis and Goadrich (2006) [ICML Proceedings], and Krzanowski and Hand (2009) [ROC Curves for Continuous Data] further highlight how small datasets yield coarse ROC curves, risking AUC overestimation.

2. Need for Transparent Error Rate Reporting

Focusing solely on AUC may obscure clinically relevant details, such as false positive (FP) and false negative (FN) rates. Including the full confusion matrix (TP, TN, FP, FN) would offer a more comprehensive view of model performance, especially in small cohorts where individual errors carry greater weight [Hanczar, B., et al. (2010). *Bioinformatics*, 26(16), 1934-1940]. This breakdown reveals specific strengths and weaknesses that summary metrics might overlook, which is vital for interpreting predictive behavior.

In small datasets, where sample sizes are limited (e.g., tens or hundreds of instances rather than thousands), relying solely on summary metrics like accuracy, precision, recall, or the Area Under the ROC Curve (AUC) can obscure critical aspects of a model's performance. The full confusion matrix, which breaks down predictions into True Positives (TP), True Negatives (TN), False Positives (FP), and False Negatives (FN), offers a granular view of how the model behaves across all classes, revealing patterns that aggregated scores might mask. This is particularly vital in small validation cohorts, where statistical noise, class imbalance, or overfitting can disproportionately skew results. For instance, a high accuracy might reflect a model correctly predicting a dominant class (TN) while failing on a rare but critical minority class (FN), a nuance only visible through the matrix. By examining all four components, researchers can assess not just overall performance but also specific

error types—crucial for applications like medical diagnostics or fraud detection, where false negatives or false positives carry different costs.

In small datasets, the limited number of samples amplifies the impact of individual misclassifications, making the confusion matrix a more reliable tool for understanding predictive behavior than metrics derived from it. With fewer data points, each TP, TN, FP, or FN contributes a larger relative weight to the overall evaluation, and random variations or biases in the data can lead to misleading conclusions if only summary statistics are considered. The matrix provides a transparent, comprehensive snapshot, enabling researchers to compute multiple metrics (e.g., sensitivity, specificity, positive predictive value) and weigh their trade-offs explicitly, rather than relying on a single, potentially deceptive number.

- Powers (2011) [Journal of Machine Learning Technologies], Kohavi (1995) [IJCAI Proceedings], Sokolova and Lapalme (2009) [Information Processing & Management], Provost et al. (1998) [ICML Proceedings], Japkowicz and Shah (2011) [Evaluating Learning Algorithms], Kuhn and Johnson (2013) [Applied Predictive Modeling], and Flach (2012) [Machine Learning] all advocate the confusion matrix's role in small datasets for its detailed, reliable insights.

Recommendations for Improvement

1. Abstract/Discussion: Clearly state the validation cohort sizes (n=65–68) alongside AUC values to contextualize their interpretation.
2. Results: Consider adding full confusion matrices for the R4RA validation to enrich the performance narrative.

These adjustments would align the manuscript with best practices in statistical reporting, enhance result interpretability, and adhere to the TRIPOD guidelines for transparent prediction modeling.

(Remarks on code availability)

The authors did not provide any scripts for review or public access.

Reviewer #2

(Remarks to the Author)

no further comments

(Remarks on code availability)

Reviewer #3

(Remarks to the Author)

The authors currently provide data generated by RNAseq of synovial biopsies obtained during the STRAP trial. Patients underwent synovial biopsies and were subsequently randomized into three treatment groups (Eta/Toc/Rtx) looking for responders and non-responders to the individual drugs.

By studying DEGs and applying deconvolution algorithms with RNAseq datasets of responders and non-responders in the different groups, the authors identify genes up- and downregulated as well as cell types that are predicted to be enriched in responders/non-responders (partially overlapping in treatment groups).

Moreover, machine-learning algorithms are successfully applied, tested and validated using these datasets and a different treatment cohort (R4RA).

General comments:

Overall, the manuscript is interesting and provides confirmatory insights into the molecular heterogeneity of RA patients and its impact on treatment responses in these patients. The work adds to the previous work of the group, which has applied similar approaches during the R4RA trial before and thus provides a proof of principle for precision medicine in rheumatology. The use of two datasets from two independent trials and the possibility to validate the developed algorithms renders this manuscript an important piece of work.

(Remarks on code availability)

Reviewer #4

(Remarks to the Author)

The Manuscript by Lewis et al has been significantly improved compared with the original version, particularly regarding clarity of the results and discussion sections as well as some methodological matters. However, the main issues still remain unresolved. While the authors disagree with this reviewer on several aspects discussed before, it is a pleasure to provide some further support for the previous comments in order to suggest a next step of improvement to the authors, which is the aim of this review. Just to make it clear: this reviewer is not an opponent of this study and is full of esteem of previous work published by the authors, but as long as the results do not meet the validity that is needed to appreciate them fully,

comments will be made and critique raised.

1. In their response to the reviewer's comment on RF and ACPA, the authors state: "We respectfully disagree...about the usefulness of RF and anti-CCP for predicting response to biologics," and then cite several papers. The publications by Chatzidionysiou and by Der Keyser come from registries with all the biases that registries have (such as bias by indication) – the authors will know quite well that registries were established to assess safety and not efficacy. Also, in contrast to what the authors suggest, the study by Lal et al which looked at clinical trial data showed that across all autoantibody subgroups evaluated the autoantibody positive patients had a much better response than the negative ones, with the reverse seen for placebo and, therefore, the difference between active and placebo arms is even more pronounced. The study clearly concludes: "The presence of any RF isotype and/or IgG anti-CCP autoantibodies together with an elevated CRP level identifies a subgroup of patients with RA in whom the benefit of rituximab treatment may be enhanced." Consequently, the EMA's SMPC for rituximab contains the following statement: "Patients seropositive to Rheumatoid Factor (RF) and/or anti-Cyclic Citrullinated Peptide (anti-CCP) who were treated with Mabthera in combination with methotrexate showed an enhanced response compared to patients negative to both." And: "The best responses to Mabthera are seen in those who have a positive blood test to rheumatoid factor (RF) and/or anti-cyclic citrullinated peptide (anti-CCP)." EMA would not have said this if the clinical trial data would not have led to this conclusion. Such statements are not found for tocilizumab or etanercept. Thus, the authors' interpretation of these data is incorrect and since the authors' study was not stratified for RF or ACPA (nor for high versus low levels of these autoantibodies), this remains an open issue and should be done, especially with respect to the response to rituximab.

2. The authors also disagree with the reviewer on the use of the DAS28. The reviewer would not hesitate a minute to concur with the authors on their disagreement, if that is in line with the available data base. However, the data simply speaks against their contention. As has been shown over and over again, the DAS28 is heavily weighted on acute phase reactants. Let us just look at one study out of many, a biomarker study: Fleischmann et al clearly showed that a biomarker test which is heavily weighted on acute phase reactants led to totally different results for abatacept, a drug that does not affect APRs directly, versus an anti-TNF, despite almost identical clinical and radiographic outcomes (Ann Rheum Dis 2023;82(6):773-87. doi: 10.1136/ard-2022-222784) – what proof do the authors still need to learn that this is not the correct endpoint for a study that compares rituximab vs etanercept let alone tocilizumab? If the best ML models depend on using the DAS28, then one simply cannot apply these models; needless to say that this model was best for...etanercept and tocilizumab, as to be anticipated!

3. The authors' argument that the ACR response cannot be applied, because it requires use of an extended joint count as opposed to the DAS28 is, again, simply incorrect; Felson et al have clearly stated that reduced joint counts can be used to calculate the ACR response rates: "The ACR committee regards validated joint counts with as few as 28 joints equally acceptable in RA clinical trials" (Arthritis Rheum 1994;37:464-465). The additional argument of the authors that they applied ML for multiple endpoints is incredibly circular, since all these endpoints, including the EULAR response criteria, depend on the DAS28. Circularity of reasoning as well as of data assessments and interpretation should be avoided, as it is not appropriate to validate data using the same methodology. Of course, one can try to find various explanations, but the one issue remains: results must hold up across outcomes. If data are developed using a low disease activity parameter by one measure, such as DAS28, one should be able to validate these results with a low disease activity parameter by another measure, be it an ACR70 or, as suggested by EMA in their guideline for RA trials (CPMP/EWP/556/95 Rev. 2), by CDAI or SDAI. If this is not the case, then the correlation may have less to do with reliability than with the application of an unreliable measure. Trying many outcomes until one finds the one that fits one's data (see section on machine learning and some answers to reviewer comments) is a questionable approach, especially if not sufficiently corrected for statistically and validated by other instruments.

4. What the authors do is truly highly desirable, namely to find means to predict response to different therapies. To this end validation of data is of utmost importance, since else the data stand only for a given set of patients at a given point in time. Therefore, as mentioned before, validation of the current data set when compared with the previous one (R4RA) is needed to understand if one can generalize the findings, at least partly. The authors provide nice ROCs comparing STRAP with R4RA data in Fig. 5. However, the data are based on the DAS28 ML-model, which depends on an invalid endpoint for drugs that target the APR directly, as stated above. Of course, one could potentially obviate this limitation by showing the expression of the same or broadly overlapping genes across the two trials. While the authors provide an explanatory Figure (4 d) regarding overlapping and non-overlapping gene clusters in the two independent cohorts at baseline in general, the signature and module patterns related to response and non-response are totally different between the plots shown in Fig 1 of the present study and those shown in Fig 2 of the R4RA study (Nat Med. 2022). This is not addressed.

Finally, regarding the prediction of B-cell gene modules for the etanercept response, the authors provide excellent arguments by pointing toward the involvement of macrophages in B-cell-rich RA synovial membranes. However, this does not sufficiently explain the difference to the tocilizumab-signatures, since IL-6 is as much a B-cell growth factor (originally described as BSF, B-cell stimulatory factor, by Kishimoto et al) as it is a macrophage- (and B-cell)-produced cytokine. Moreover, since TNF induces IL-6 (hence its rapid effect on APRs), it is not easy to understand the difference between tocilizumab and etanercept but not rituximab and etanercept in this study. But: data are data and if the authors can validate their ML-model using different instruments for low disease activity and show that the same genes are expressed in responders and non-responders across trials, then the paper would fulfil all requirements for an excellent manuscript.

(Remarks on code availability)

REVIEWERS' COMMENTS

Reviewer #1 (Remarks to the Author):

The manuscript presents a compelling and potentially impactful study exploring the prediction of treatment responses to biologic therapies in Rheumatoid Arthritis. Leveraging RNA-sequencing data from the STRAP trial alongside machine learning to develop predictive models is a notable strength of this work. Additionally, the validation in an independent cohort and the adaptation of findings into an nCounter panel for potential clinical application are highly valuable contributions. That said, addressing certain statistical and methodological aspects could further enhance the rigor, reproducibility, and overall clarity of the findings.

Availability of Code and Scripts

The analysis of RNA-Seq data and the development of machine learning models offer significant insights into predicting treatment responses in RA. However, the absence of publicly accessible code and scripts presents a challenge to independently verifying the results and fully understanding the methodology employed.

1. Supporting Reproducibility Standards

Reproducibility is a fundamental pillar of scientific research. Many journals and funding bodies increasingly advocate for or require code sharing to allow independent validation and build upon published work [Peng, R. D. (2011). *Science*, 334(6060), 1226-1227; Sandve, G. K., et al. (2013). *PLoS Computational Biology*, 9(10), e1003282]. The FAIR principles further underscore the value of making data and code findable, accessible, interoperable, and reusable to benefit the broader research community [Wilkinson, M. D., et al. (2016). *Scientific Data*, 3, 160018].

2. Enhancing Methodological Clarity

Providing analysis scripts would greatly clarify several key steps that remain somewhat ambiguous in the manuscript:

- RNA-Seq Data Analysis: Scripts could specify quality control metrics, trimming parameters, alignment tools (e.g., STAR, HISAT2) with versions, and the normalization approach in DESeq2 (e.g., relative log expression or trimmed mean of M-values).
- Batch Effect Management: Code would indicate whether batch effects (e.g., from sequencing runs or biopsy sites) were addressed using methods like ComBat or SVA, and how these adjustments were incorporated into DESeq2.
- Differential Gene Expression: Scripts would detail the DESeq2 model formula, including covariates (e.g., age, sex, disease duration) beyond biopsy joint size, as well as gene filtering and p-value adjustment criteria.
- Machine Learning Models: Code would elucidate feature selection (e.g., t-test or ANOVA thresholds, LASSO tuning), algorithm details (e.g., random forest hyperparameters), and the implementation of nested cross-validation and SMOTE oversampling.
- Independent Validation (R4RA): Scripts would confirm consistent processing between STRAP and R4RA cohorts, including normalization and batch effect alignment.

3. Fostering Transparency and Confidence

Sharing scripts would enhance transparency, allowing readers to evaluate the appropriateness of the methods and identify any potential limitations, thereby strengthening trust in the study's conclusions.

Suggestion

To address this, I kindly recommend that the authors consider depositing all code and scripts used for data processing, statistical analysis, and model development in a public repository (e.g., GitHub, GitLab). The repository could include:

- Clearly documented scripts with step-by-step explanations.
- Instructions for replicating the analysis from raw data to final results.
- Software version details (e.g., R, Python, DESeq2, scikit-learn).

This step would align the study with contemporary standards for reproducible research and significantly bolster its credibility.

If sharing raw data is restricted due to privacy concerns, the authors might consider:

- Providing Markdown files or Jupyter notebooks with:
- Complete analysis scripts.
- Code to load intermediate data (e.g., normalized expression matrices, DESeq2 results, feature-selected datasets), shareable without raw data.
- Guidance on running the scripts to replicate key findings from these intermediates.

This approach would balance transparency with patient confidentiality, enabling reviewers and researchers to follow the workflow without accessing sensitive data.

Statistical Observations

The manuscript reports AUC values of 0.786 (rituximab, n=65) and 0.713 (tocilizumab, n=68) for the R4RA validation cohort, which are promising results. However, the Abstract and Discussion sections could benefit from explicitly noting the modest sample sizes to provide context and avoid overstating the reliability of these estimates.

1. Impact of Small Validation Cohorts on AUC Stability

AUC values derived from small cohorts ($n < 100$) can exhibit considerable variability and may overestimate true performance due to limited data points shaping the ROC curve [Hanley, J. A., & McNeil, B. J. (1982). *Radiology*, 143(1), 29-36]. In smaller datasets, the ROC curve may appear "step-like" because fewer distinct prediction thresholds are available, potentially inflating the AUC [Lobo, J. M., et al. (2008). *Global Ecology and Biogeography*, 17(2), 145-151]. This effect stems from the discrete jumps in True Positive Rate (TPR) and False Positive Rate (FPR) with fewer samples, rather than the smooth curve typical of larger datasets, which can exaggerate class separation if overfitting or chance alignments occur.

In small datasets, the AUC might be inflated because the limited thresholds can exaggerate the separation between classes, especially if the model overfits or if random chance aligns the few data points favorably. The AUC is calculated as the integral under the ROC curve, and with fewer steps, the area can appear artificially high if the steps occur at points that maximize TPR while minimizing FPR, even if this doesn't reflect true generalizability. This inflation is particularly problematic because AUC is often interpreted as the probability that a randomly chosen positive instance is ranked higher than a negative one—a metric that assumes a robust sampling of the underlying distribution, which small datasets can't reliably provide.

- Fawcett (2006) [Pattern Recognition Letters] notes that ROC granularity depends on sample size, with fewer points creating a step-like curve.
- Hanley and McNeil (1982) emphasize AUC's sample-size dependency.
- Berrar and Flach (2012) [Briefings in Bioinformatics], Davis and Goadrich (2006) [ICML Proceedings], and Krzanowski and Hand (2009) [ROC Curves for Continuous Data] further highlight how small datasets yield coarse ROC curves, risking AUC overestimation.

2. Need for Transparent Error Rate Reporting

Focusing solely on AUC may obscure clinically relevant details, such as false positive (FP) and false negative (FN) rates. Including the full confusion matrix (TP, TN, FP, FN) would offer a more comprehensive view of model performance, especially in small cohorts where individual errors carry greater weight [Hanczar, B., et al. (2010). *Bioinformatics*, 26(16), 1934-1940]. This breakdown reveals specific strengths and weaknesses that summary metrics might overlook, which is vital for interpreting predictive behavior.

In small datasets, where sample sizes are limited (e.g., tens or hundreds of instances rather than thousands), relying solely on summary metrics like accuracy, precision, recall, or the Area Under the ROC Curve (AUC) can obscure critical aspects of a model's performance. The full confusion matrix, which breaks down predictions into True Positives (TP), True Negatives (TN), False Positives (FP), and False Negatives (FN), offers a granular view of how the model behaves across all classes, revealing patterns that aggregated scores might mask. This is particularly vital in small validation cohorts, where statistical noise, class imbalance, or overfitting can disproportionately skew results. For instance, a high accuracy might reflect a model correctly predicting a dominant class (TN) while failing on a rare but critical minority class (FN), a nuance only visible through the matrix. By examining all four components, researchers can assess not just overall performance but also specific error types—crucial for applications like medical diagnostics or fraud detection, where false negatives or false positives carry different costs.

In small datasets, the limited number of samples amplifies the impact of individual misclassifications, making the confusion matrix a more reliable tool for understanding predictive behavior than metrics derived from it. With fewer data points, each TP, TN, FP, or FN contributes a larger relative weight to the overall evaluation, and random variations or biases in the data can lead to misleading conclusions if only summary statistics are considered. The matrix provides a transparent, comprehensive snapshot, enabling researchers to compute multiple metrics (e.g., sensitivity, specificity, positive predictive value) and weigh their trade-offs explicitly, rather than relying on a single, potentially deceptive number.

- Powers (2011) [Journal of Machine Learning Technologies], Kohavi (1995) [IJCAI Proceedings], Sokolova and Lapalme (2009) [Information Processing & Management], Provost et al. (1998) [ICML Proceedings], Japkowicz and Shah (2011) [Evaluating Learning Algorithms], Kuhn and Johnson (2013) [Applied Predictive Modeling], and Flach (2012) [Machine Learning] all advocate the confusion matrix's role in small datasets for its detailed, reliable insights.

Recommendations for Improvement

1. Abstract/Discussion: Clearly state the validation cohort sizes (n=65–68) alongside AUC values to contextualize their interpretation.
2. Results: Consider adding full confusion matrices for the R4RA validation to enrich the performance narrative.

These adjustments would align the manuscript with best practices in statistical reporting, enhance result interpretability, and adhere to the TRIPOD guidelines for transparent prediction modeling.

Reviewer #1 (Remarks on code availability):

The authors did not provide any scripts for review or public access.

We thank the statistical reviewer for their comments. We have provided a github repo with full scripts covering processing of the raw RNA-Seq data and creation of the RNA-Seq count

matrix through to differential expression analysis, gene module analysis, clustering, heatmaps, selection of optimal machine learning models, fitting and testing of performance of models through repeated nested CV.

Software version and package version numbers are provided in detail in the Methods.

We have released the embargo on the RNA-Seq data. It is fully available and public through ArrayExpress.

We accept that AUC on small sample sizes can be variable. For this reason we did not simply perform a single round of nested CV, but performed nested CV with 25 repeats (folds randomised each time). Supplementary Table 4 shows performance metrics which not only include AUC, but in addition include area under precision recall curve (AUC.PR), accuracy, balanced accuracy, F1 score and Matthew's correlation coefficient. Also, for all of these metrics estimates of the variability of performance across 25 repeats is shown in the table. Also, in figure 5b we show the actual results of the nested CV across each of the 25 repeats (each point represents a repeat). This shows the variability in AUC across repeats for each model type.

We have added a sentence to the discussion to highlight the limitation of the sample size in the AUC estimate as follows:

“Model performance metrics such as AUC can be affected by small sample size, so we have provided full confusion matrices for all models (Supplementary Tables 5-7) and estimates of the variability of AUC and performance metrics across 25 repeats of the nested CV.”

We have provided a comprehensive github repo with annotated scripts for the entire analysis. The nestedcv package which is the main engine of the machine learning modelling is already fully public and open source and can be installed directly from the CRAN R repository.

Reviewer #2 (Remarks to the Author):

no further comments

Reviewer #3 (Remarks to the Author):

The authors currently provide data generated by RNAseq of synovial biopsies obtained during the STRAP trial. Patients underwent synovial biopsies and were subsequently randomized into three treatment groups (Eta/Toc/Rtx) looking for responders and non-responders to the individual drugs.

By studying DEGs and applying deconvolution algorithms with RNAseq datasets of responders and non-responders in the different groups, the authors identify genes up- and downregulated as well as cell types that are predicted to be enriched in responders/non-responders (partially overlapping in treatment groups).

Moreover, machine-learning algorithms are successfully applied, tested and validated using these datasets and a different treatment cohort (R4RA).

General comments:

Overall, the manuscript is interesting and provides confirmatory insights into the molecular heterogeneity of RA patients and its impact on treatment responses in these patients. The work adds to the previous work of the group, which has applied similar approaches during

the R4RA trial before and thus provides a proof of principle for precision medicine in rheumatology. The use of two datasets from two independent trials and the possibility to validate the developed algorithms renders this manuscript an important piece of work.

We would like to thank the reviewer for the constructive and supportive comments.

Reviewer #4 (Remarks to the Author):

The Manuscript by Lewis et al has been significantly improved compared with the original version, particularly regarding clarity of the results and discussion sections as well as some methodological matters. However, the main issues still remain unresolved. While the authors disagree with this reviewer on several aspects discussed before, it is a pleasure to provide some further support for the previous comments in order to suggest a next step of improvement to the authors, which is the aim of this review. Just to make it clear: this reviewer is not an opponent of this study and is full of esteem of previous work published by the authors, but as long as the results do not meet the validity that is needed to appreciate them fully, comments will be made and critique raised.

1. In their response to the reviewer's comment on RF and ACPA, the authors state: "We respectfully disagree...about the usefulness of RF and anti-CCP for predicting response to biologics," and then cite several papers. The publications by Chatzidionysiou and by Der Keyser come from registries with all the biases that registries have (such as bias by indication) – the authors will know quite well that registries were established to assess safety and not efficacy. Also, in contrast to what the authors suggest, the study by Lal et al which looked at clinical trial data showed that across all autoantibody subgroups evaluated the autoantibody positive patients had a much better response than the negative ones, with the reverse seen for placebo and, therefore, the difference between active and placebo arms is even more pronounced. The study clearly concludes: "The presence of any RF isotype and/or IgG anti-CCP autoantibodies together with an elevated CRP level identifies a subgroup of patients with RA in whom the benefit of rituximab treatment may be enhanced."

Consequently, the EMA's SMPC for rituximab contains the following statement: "Patients seropositive to Rheumatoid Factor (RF) and/or anti-Cyclic Citrullinated Peptide (anti-CCP) who were treated with Mabthera in combination with methotrexate showed an enhanced response compared to patients negative to both." And: "The best responses to Mabthera are seen in those who have a positive blood test to rheumatoid factor (RF) and/or anti-cyclic citrullinated peptide (anti-CCP)." EMA would not have said this if the clinical trial data would not have led to this conclusion. Such statements are not found for tocilizumab or etanercept. Thus, the authors' interpretation of these data is incorrect and since the authors' study was not stratified for RF or ACPA (nor for high versus low levels of these autoantibodies), this remains an open issue and should be done, especially with respect to the response to rituximab.

We have already provided the data requested by the reviewer. We show response rates in both STRAP and STRAP combined with R4RA trials comparing response using primary and secondary endpoints between ACPA positive and RF positive patients in Supplementary Table 8. We also already provided data showing that CCP or RF titre was not a strong predictor of response in STRAP and R4RA (Supplementary Fig 6) for any drug.

In light of the comments of the reviewer we have modified the discussion to include more information and detail about these results in STRAP and R4RA.

The reviewer quotes the paper by Lal et al which was an analysis of 2 major RCT. This paper as we stated in our previous rebuttal has been superseded by a meta-analysis of 4 RCT by Isaacs et al (2012) which included the data from the 2 RCT (SERENE, REFLEX)

analysed by Lal et al. Thus, Isaacs et al includes data from n=2177 patients (n=1466 rituximab) whereas Lal analysed data from n=1008 (n=635 rituximab). We have discussed this paper (Isaacs et al – ref 44 below) in the Discussion, as well as citing Lal et al (ref 43 below), reported below for easy reference:

“Multiple studies have also investigated whether anti-CCP or rheumatoid factor titre or seropositivity predict response to biologics.⁴³ Some of the most comprehensive studies include a meta-analysis of four key RCT which included 1416 rituximab treated patients,⁴⁴ and an observational study of 27,583 RA patients treated with four different biologics.⁴⁵ These show that seropositive patients have a tendency to modestly higher response rates when treated with rituximab or tocilizumab, but not anti-TNF inhibitors. Collectively, these studies reveal that the difference in response rates seropositive and seronegative patients is modest and that anti-CCP and/or rheumatoid factor positivity are weak predictors of response.”

We have then amended the sentences in the Discussion that follow the above section of text:

“This is consistent with our own analysis of STRAP and R4RA which shows that CCP and RF titre as a continuous variable was not a strong predictor of response (Supplementary Fig. 6). However, RF and CCP positivity showed a modest association with increased response to rituximab and etanercept in STRAP (Supplementary Table 8), but the association with rituximab response was no longer significant when STRAP and R4RA were combined. No significant association was observed for tocilizumab in STRAP or in R4RA.”

We have added the following text to the Results section (p.12-13):

“Confusion matrices (Supplementary Table 7) showed that the prediction models enriched the delineation of responders and non-responders: individuals predicted to be tocilizumab responders showed an enriched response rate of 75% (61/81) compared to 32% (17/53) response in predicted non-responders. Similarly, rituximab predicted responders showed an actual response rate of 76% (16/21) compared to 18% (21/119) response in the predicted non-response group. In contrast in STRAP & R4RA combined, RF or CCP positive patients showed no difference in response rates compared to seronegative patients following treatment with tocilizumab (57% vs 61-63%) and only a moderate increase in response rate with rituximab (28-30% vs 17-19%) (Supplementary Table 8). Thus, the new prediction models provide additional benefit above and beyond simple seropositivity for predicting response to biologic therapies.”

This clarifies the benefits of the new prediction models in our study above and beyond the modest predictive ability of RF/CCP seropositivity.

2. The authors also disagree with the reviewer on the use of the DAS28. The reviewer would not hesitate a minute to concur with the authors on their disagreement, if that is in line with the available data base. However, the data simply speaks against their contention. As has been shown over and over again, the DAS28 is heavily weighted on acute phase reactants. Let us just look at one study out of many, a biomarker study: Fleischmann et al clearly showed that a biomarker test which is heavily weighted on acute phase reactants led to totally different results for abatacept, a drug that does not affect APRs directly, versus an anti-TNF, despite almost identical clinical and radiographic outcomes (Ann Rheum Dis 2023;82(6):773-87. doi: 10.1136/ard-2022-222784) – what proof do the authors still need to learn that this is not the correct endpoint for a study that compares rituximab vs etanercept let alone tocilizumab? If the best ML models depend on using the DAS28, then one simply

cannot apply these models; needless to say that this model was best for...etanercept and tocilizumab, as to be anticipated!

We have added text to the Discussion to highlight the reviewer's concern regarding the DAS28. We have written the following:

The CDAI score is preferred for comparing anti-IL-6 therapeutics against other drugs such as placebo or standard of care, due to the absence of acute phase reactants such as CRP which are driven down by its mechanism of action and thus lead to artificially higher response rates according to measures such as DAS28-CRP.⁴⁶ But in our case we are comparing tocilizumab responders vs non-responders, i.e. we are comparing tocilizumab against itself, so the tocilizumab non-responders still represent a valid non-response group which has less response to tocilizumab compared to the tocilizumab responder group. We attempted to fit models based on CDAI 50% response, but were unable to develop reliable models to predict this outcome measure for any of the three trials drugs. Our interpretation is that CDAI, which relies on both patient and clinician reported measures and includes no biological measures, is more subjective. However we were able to develop models which could predict response defined as DAS28-ESR < 3.2. The fact that the DAS28 is less subjective in that it does not include a clinician reported measure, plus the inclusion of ESR as an objective biological measure of systemic inflammation in the endpoint might have aided the prediction modelling. The optimal choice of clinical response metrics is controversial and may vary depending on the drug being studied and the scientific question being asked – no response measure is perfect for all drugs and all clinical scenarios. Composite scores and subcomponents show major differences in correlation with functional outcomes such as future joint damage and function/disability.⁴⁷ Studies have shown that acute phase reactants and swollen joint counts are the dominant predictors of radiographic joint damage.^{48,49} Thus the development of models which predict endpoints which include acute phase reactants may have a long term advantage for patients. Overall however, we fully accept that in an ideal world it would be preferable to predict CDAI response, especially for anti-IL6 therapeutics, but this might require substantially larger cohorts due to the higher subjectivity of the CDAI measure.

3. The authors' argument that the ACR response cannot be applied, because it requires use of an extended joint count as opposed to the DAS28 is, again, simply incorrect; Felson et al have clearly stated that reduced joint counts can be used to calculate the ACR response rates: "The ACR committee regards validated joint counts with as few as 28 joints equally acceptable in RA clinical trials" (Arthritis Rheum 1994;37:464-465). The additional argument of the authors that they applied ML for multiple endpoints is incredibly circular, since all these endpoints, including the EULAR response criteria, depend on the DAS28. Circularity of reasoning as well as of data assessments and interpretation should be avoided, as it is not appropriate to validate data using the same methodology. Of course, one can try to find various explanations, but the one issue remains: results must hold up across outcomes. If data are developed using a low disease activity parameter by one measure, such as DAS28, one should be able to validate these results with a low disease activity parameter by another measure, be it an ACR70 or, as suggested by EMA in their guideline for RA trials (CPMP/EWP/556/95 Rev. 2), by CDAI or SDAI. If this is not the case, then the correlation may have less to do with reliability than with the application of an unreliable measure. Trying many outcomes until one finds the one that fits one's data (see section on machine learning and some answers to reviewer comments) is a questionable approach, especially if not sufficiently corrected for statistically and validated by other instruments.

We understand the reviewer's argument, but we remain of a different opinion on this point, as the presumption by the reviewer that "*results must hold up across outcomes*" cannot be correct since different response outcomes correlate with different biological endpoints (Smolen JS 2014, Clin Exp Rheum 32 (Suppl. 85): S75-S79; Navarro-Compan V 2015,

Rheumatology 54:994-1007). For example, acute phase reactants and swollen joints correlate much more strongly with long term structural joint damage as proven by x-rays. Therefore, endpoints such as DAS28 or DAS2c, which weight acute phase reactants and swollen joints more highly, will be more strongly correlated with x-ray damage than measures that do not include acute phase reactants such as CDAI. Since different patients are categorised as responder vs non-responder according to different endpoints, with some patients switching from responder using a milder metric to non-responder when using a more stringent metric, it is no surprise that different results are found when differential gene expression is performed, as such expression is linked to different biological process, measured slightly differently by different outcome measures.

4. What the authors do is truly highly desirable, namely to find means to predict response to different therapies. To this end validation of data is of utmost importance, since else the data stand only for a given set of patients at a given point in time. Therefore, as mentioned before, validation of the current data set when compared with the previous one (R4RA) is needed to understand if one can generalize the findings, at least partly. The authors provide nice ROCs comparing STRAP with R4RA data in Fig. 5. However, the data are based on the DAS28 ML-model, which depends on an invalid endpoint for drugs that target the APR directly, as stated above. Of course, one could potentially obviate this limitation by showing the expression of the same or broadly overlapping genes across the two trials. While the authors provide an explanatory Figure (4 d) regarding overlapping and non-overlapping gene clusters in the two independent cohorts at baseline in general, the signature and module patterns related to response and non-response are totally different between the plots shown in Fig 1 of the present study and those shown in Fig 2 of the R4RA study (Nat Med. 2022). This is not addressed.

While, again, we understand the reviewer's argument, we disagree with the premise that "the DAS28 ML-model depends on an invalid endpoint for drugs that target the APR directly", since a change in such outcome measure is likely to more truthfully reflect the biology of response, rather than patient perception. In addition, it is not surprising that some genes are different between STRAP and R4RA patients, since these are 2 different cohorts at different stages of the disease: R4RA includes late stage disease with patients with greater joint damage at baseline entry to the trial and notoriously more difficult to respond; STRAP includes patients who have only failed csDMARDs such as methotrexate and thus earlier in their disease course. Thus, response associated genes may well differ between the cohorts. The reviewer seems to ignore the fact that two of the response models (rituximab & tocilizumab models) built in STRAP were successfully validated when applied to RNA-Seq data from R4RA. This clearly demonstrates that while some genes differ at different disease stages, others overlap at least in terms of predicting response and that machine learning models which combine genes can predict response across 2 independent trials.

Finally, regarding the prediction of B-cell gene modules for the etanercept response, the authors provide excellent arguments by pointing toward the involvement of macrophages in B-cell-rich RA synovial membranes. However, this does not sufficiently explain the difference to the tocilizumab-signatures, since IL-6 is as much a B-cell growth factor (originally described as BSF, B-cell stimulatory factor, by Kishimoto et al) as it is a macrophage- (and B-cell)-produced cytokine. Moreover, since TNF induces IL-6 (hence its rapid effect on APRs), it is not easy to understand the difference between tocilizumab and etanercept but not rituximab and etanercept in this study. But: data are data and if the authors can validate their ML-model using different instruments for low disease activity and show that the same genes are expressed in responders and non-responders across trials, then the paper would fulfil all requirements for an excellent manuscript.

We respectfully disagree with the reviewer, as it would appear that the reviewer misunderstands how machine learning models work. Each gene is not a simple biomarker

such as CRP. Machine learning models are by their nature more complex and play off multiple genes against each other as well as against the incorporated clinical variables. Thus, looking at whether genes are higher or lower in responders vs non-responders in different cohorts is a highly simplistic way of viewing these prediction models. If it was as simple as finding a single gene which was reliably higher in responders across multiple cohorts, a model including 15-30 genes would not be required. It should be possible to simply measure that one gene in every patient and that one single gene would be a sufficient biomarker on its own. The fact that this is not possible and has not already been found by other studies already shows that a more complex, nuanced approach such as a machine learning model is required.

We agree with the reviewer when she/he says “data are data”. Notably we have already presented the data from both studies in full including tables of differentially expressed genes between responders and non-responders across each of the study drugs in STRAP and in the previous publication in *Nature medicine* for R4RA. STRAP and R4RA are the largest trials to date which include bulk RNA-Seq of synovium in patients treated within properly randomised controlled trials. There are no other trials in which synovial biopsy RNA-Seq has been performed which could be included to enlarge the current analysis.